# *Patterns in Woody Vegetation Structure across African Savannas*

Christoffer R. Axelsson[1] and Niall P. Hanan[2]

[1]Geospatial Sciences Center of Excellence, South Dakota State University, Brookings, SD, USA

[2]Plant and Environmental Sciences, New Mexico State University, Las Cruces, NM, USA

*Correspondence to*: Christoffer R. Axelsson (christoffer.axelsson@sdstate.edu)

**Abstract.** Vegetation structure in water-limited systems is to a large degree controlled by ecohydrological processes, including mean annual precipitation (MAP) modulated by the characteristics of precipitation and geomorphology that collectively determine how rainfall is distributed vertically into soils or horizontally in the landscape. We anticipate that woody canopy cover, crown density, crown size, and the level of spatial aggregation among woody plants in the landscape, will vary across environmental gradients. A high level of woody plant aggregation is most distinct in periodic vegetation patterns (PVPs), which emerge as a result of ecohydrological processes such as runoff generation and increased infiltration close to plants. Similar, albeit weaker, forces may influence the spatial distribution of woody plants elsewhere in savannas. Exploring these trends can extend our knowledge of how semi-arid vegetation structure is constrained by rainfall regime, soil type, topography, and disturbance processes such as fire. Using high spatial resolution imagery, a flexible classification framework, and a crown delineation method, we extracted woody vegetation properties from 876 sites spread over African savannas. At each site, we estimated woody cover, mean crown size, crown density, and the degree of aggregation among woody plants. This enabled us to elucidate the effects of rainfall regimes (MAP and seasonality), soil texture, slope, and fire frequency on woody vegetation properties. We found that previously documented increases in woody cover with rainfall is more consistently a result of increasing crown size than increasing density of woody plants. Along a gradient of mean annual precipitation from the driest (<200 mm/yr) to the wettest (1200-1400 mm/yr) end, mean estimates of crown size, crown density, and woody cover increased by 233 %, 73 %, and 491 % respectively. We also found a unimodal relationship between mean crown size and sand content suggesting that maximal savanna tree-sizes do not occur in either coarse sands or heavy clays. When examining the occurrence of PVPs, we found that the same factors that contribute to the formation of PVPs also correlate with higher levels of woody plant aggregation elsewhere in savannas and that rainfall seasonality plays a key role for the underlying processes.

## 1 Introduction

African savannas are complex tree-grass systems controlled by combinations of climate, soil, and disturbance processes such as fire and herbivory (Sankaran et al., 2008). In dry savannas, water availability determines the establishment, growth and survival of plants and competitive plant traits are often of a water saving nature (Chesson et al., 2004; Pillay & Ward, 2014). Abiotic environmental factors, such as the rainfall regime, soil type, and topography, impact ecohydrological processes by controlling infiltration rates, runoff generation, and available water capacity, which in turn impact the growth and survival of woody plants in the landscape (Ludwig et al., 2005). Climate,

both rainfall patterns and temperatures, could change in many parts of Africa (Gan et al., 2016), and the effect on vegetation will depend on how those pressures interact with other abiotic and biotic factors. In addition to ecohydrological factors, savannas are heavily influenced by the frequency and intensity of fires (Bond, 2008), as well as herbivore regimes (Sankaran et al., 2008), which often combine to suppress woody cover to levels well below its climatic potential (Sankaran et al., 2005). A thorough understanding of the underlying processes that influence savanna vegetation structure is key to assessing the future resilience and productivity of these ecosystems.

Across environmental gradients we expect to see variation in woody vegetation properties, including individual-level characteristics (mean crown size) and population-level characteristics (crown density, woody cover and the spatial distribution of plants in the landscape). Woody cover is fundamentally a function of crown sizes and crown density and by studying these components individually, it is possible to attain important insight into the function of ecosystems and what ecosystem services they provide. Two landscapes with similar woody cover but different sizes of individual trees will sequester different amounts of carbon (Shackleton & Scholes, 2011), harbor different fauna (Riginos & Grace, 2008), and differ in biogeochemical dynamics (Veldhuis, Hulshof, et al., 2016). The level of spatial aggregation among woody plants can help us understand facilitative and competetive processes determining survival of seedlings and saplings. Woody plants increase water infiltration and local accumulation of soil and nutrient resources, as well as altering sub-canopy microclimates (Barbier et al., 2014; Dohn et al., 2016; Gómez-Aparicio et al., 2008). These short-range facilitative effects usually operate at spatial scales of a few meters, but may increase the degree of aggregation among woody plants at larger scales (Scanlon et al., 2007; Xu et al., 2015). Overland flows of water can be especially effective at redistributing resources over longer distances, in some conditions leading to the emergence of periodic vegetation patterns (PVPs; Rietkerk & van de Koppel, 2008; Valentin et al., 1999). Contrasting infiltration rates between bare and vegetated patches lead to redistribution of water and soil resources which reinforces an organized pattern. While soil texture type has been weakly associated with the occurrence of PVPs (Deblauwe et al., 2008), the impervious conditions of the bare patches are generally caused by shallow soil depths, hardpans, or soil crusts (McDonald et al., 2009). On flat ground, PVPs take the form of spotted, labyrinthine, or gapped patterns depending on soil water availability. On a gentle slope, they develop into vegetated bands that run parallel to contour lines (Valentin et al., 1999). While PVPs have been studied extensively, their formative processes are seldom linked to ecohydrological processes in other types of savanna landscapes.

To analyze how woody cover, crown size, crown density and the spatial pattern of trees vary with environmental gradients, we need to map the landscape at the level of individual trees. Satellite-based high spatial resolution (HSR; <4 m) sensors have the necessary degree of detail for this task. Papers delineating individual trees from HSR in African savannas have shown promising results (Karlson et al., 2014; Rasmussen et al., 2011), but these studies are generally restricted to small geographical areas. In this paper we present an analysis of woody properties sampled across the diverse water-limited savannas of Africa using a combination of WorldView, Quickbird and GeoEye satellite data ($\leq$ 0.61 m resolution) from 876 sites. The woody components of the sites were classified and delineated into individual tree crowns, from which we derived estimates of mean crown size, crown density, woody cover, and the degree of

aggregation among woody plants. We then analyzed how these woody vegetation properties vary with rainfall regime (MAP and seasonality), soil texture, slope, and fire frequency using a boosted regression tree (BRTs) approach. The dataset contains sites from several areas with PVPs and we also investigated the environmental factors associated with the occurrence of highly organized periodic patterns.

**2 Data and Methodology**

Our methodological approach included a flexible classification approach based on unsupervised classification, tree crown delineation, and boosted regression tree analysis (Figure 1).

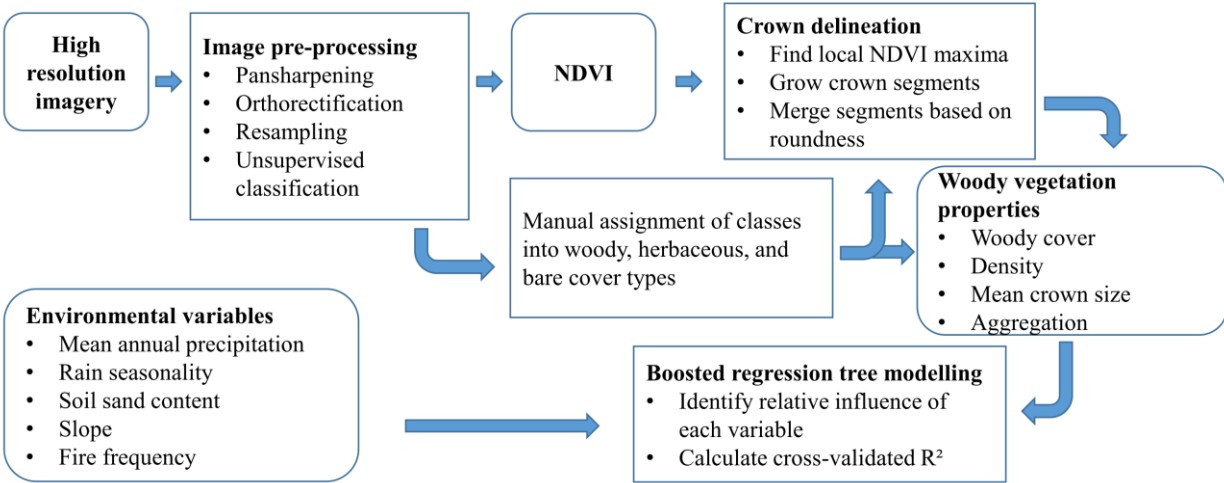

**Figure 1. Methodological workflow showing datasets (rounded boxes) and methods (square boxes) used to estimate woody vegetation structure and analyze relationships with environmental variables.**

**2.1 Satellite data and sampling strategy**

We used data from WorldView-2, WorldView-3, GeoEye-1, and Quickbird-2 satellites, with varying ground
resolutions (≤0.61 m for panchromatic data and ≤2.44 m for the multispectral bands). The sampling frame for the analysis was sub-Saharan African savannas with a minimum of anthropogenic disturbances. When acquiring data, we adopted a sampling strategy with imagery distributed across Africa in rangelands as defined by the Anthropogenic biomes product (Ellis & Ramankutty, 2008) (Figure 2), which helped us identify and avoid areas with high anthropogenic impact.Focus was on selecting recent images (2011-2016) when trees were in full leaf (generally in
mid to late growing season) and avoid areas of high human population density. The selection process was also influenced by a second study on change detection where we needed overlapping imagery from two points in time. We excluded images with view angles >25° or cloud cover >20%. Following these criteria, we acquired imagery in 48 regions, within which we sampled 240 x 240 m sites for use in the analysis. Within-image site-selection followed a systematic sampling approach and was guided by a 0.04° longitude/latitude grid which served as a base for site
locations. In some cases, however, the location of sites was adjusted to avoid areas where vegetation structure was clearly influenced by topography (rocky outcrops, streams and gullies), or anthropogenic activity (settlements, roads,

active or fallow agriculture). Sample locations influenced by topographic or anthropogenic effects were either moved to a nearby location or eliminated from the analysis. During the later classification process, we found that some sites could not be classified reliably due to either low image quality, or a lack of contrast between trees and the herbaceous

background. These sites were also eliminated. In the end, we ended up with a total of 876 sites (Figure 2).

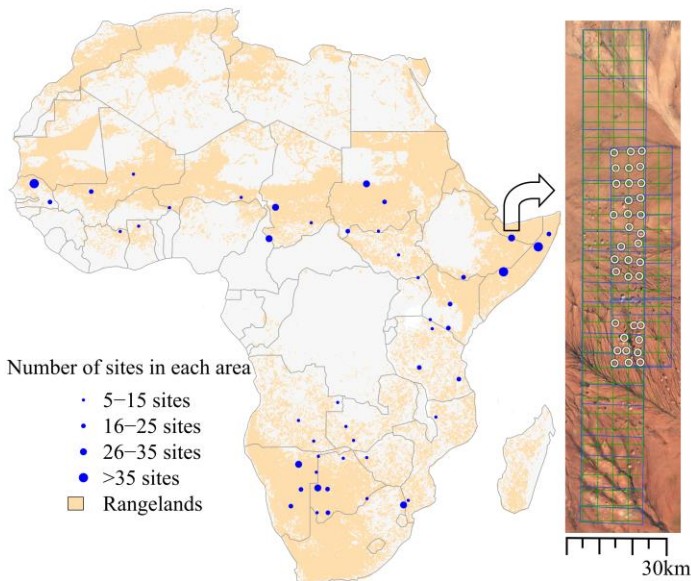

**Figure 2:  Location of the 48 study areas, containing 876 study sites, spread out over African rangelands. The rangeland areas are from the Anthropogenic biomes product (Ellis & Ramankutty, 2008), and symbol size for study areas is**

**proportional to the number of study sites in each. The map to the right shows a study area on the border between Somalia and Ethiopia and exemplifies the sampling strategy for study sites (white rings). The placement of sites was guided by a 0.04º longitude/latitude grid (green lines) in areas with overlapping older and newer satellite imagery (blue lines).**

### 2.2 Preprocessing and classification of satellite data

Once the locations of sites were established, each site was preprocessed using IDL scripts in ENVI 5.2. This included Gram-Schmidt pan-sharpening of the blue, green, red, and infrared bands, and orthorectification using embedded RPC-information and an SRTM v2 DEM (Farr et al., 2007). The orthorectified images were resampled using a nearest neighbor method to a standard 0.6 m ground resolution creating a 400 x 400 pixel (240 x 240 m) image centered over each site. We then ran unsupervised ISODATA classification on the pan-sharpened images to create 18 spectrally

different classes, which were smoothed using a kernel size of 3 pixels. Following preprocessing, the 18 spectrally distinct classes were manually assigned to woody, herbaceous and bare cover classes using a custom-built software in R. The software includes several tools to facilitate accurate and efficient classifications, including a tool to split a class into two spectrally different classes if it appears to contain more than one land cover type, and a tool to remove minor inconsistencies such as a single herbaceous pixel in the middle of a tree crown.

## 2.3 Crown delineation

After the 240 x 240 m image constituting each study site was classified into woody, herbaceous, and bare soil components, a crown delineation process was run to aggregate woody pixels into individual tree crown polygons. The method uses the classified woody layer (as the "forest mask") together with NDVI from the pansharpened imagery

and is based on the assumption that woody plants have higher NDVI at the center of the crown, where branches and leaves are dense, and declining NDVI towards the outer edges of the crown where branch and leaf density tend to be lower. The first step in the delineation process is to identify local maxima in NDVI. If the center pixel in the 3 x 3 pixel neighborhood is a maximum, it is given a unique segment ID and serves as a seed for a crown segment. The second step involves iterative growth of segments in all directions, but only to woody pixels with lower NDVI than

the neighboring segment pixel. In the third step, neighboring segments are merged if the resulting crown is rounder than both of the two neighboring segments. Since the merging criteria can be fulfilled for several segment neighbors, a segment is only merged once in each iteration and the merging order is based on the roundness of the resulting segments. Here, roundness is calculated as the area of the segment divided by the area of a minimum bounding circle. Round segments thereby get values close to one, while more complex segment forms have lower roundness. This step

is re-iterated until rounder segments cannot be formed (Figure 3). We also added a maximum crown size limit so that segments are not merged if the resulting crown is larger than the area of a circle with diameter 40 m, as trees larger than this size are very rare throughout the sampling frame. The method was implemented in C code and has several traits in common with previous delineation methods (e.g. Bunting & Lucas, 2006; Culvenor, 2002; Karlson et al., 2014; Pouliot & King, 2005) which generally were developed and tuned for a specific landscape type. The method by

Bunting & Lucas (2006) is perhaps the most similar since it also identifies segment seeds using local maxima of a vegetation index, iteratively expands to neighboring pixels, and has iterations of segment merging. That method was developed using the eCognition software and has some additional steps not included in our method, such as post-splitting of segments and the initial generation of a forest mask. In our methodology, the forest mask (woody areas) was already established using the semi-automatic approach described above.


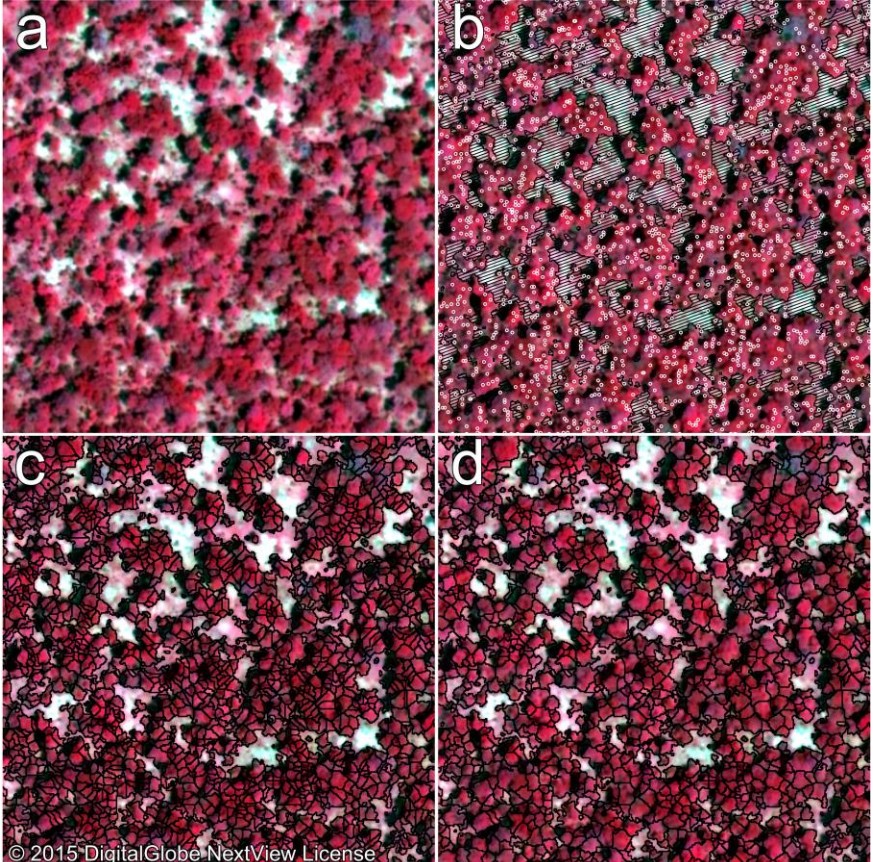

**Figure 3: Crown delineation steps for a woodland site in Zambia. (a) Pan-sharpened false-color image, (b) Local NDVI maxima as white points and the non-woody areas shown as striped polygons, (c) Crown segments before merging, and (d) and the final crown polygons following crown merging.**


The delineated crowns played an important role in this analysis because they were used for calculating crown density, crown sizes, and woody plant aggregation. Our analysis of the derived woody properties did not focus on absolute numbers but on how they vary across environmental gradients under the assumption that errors were propagated consistently over space. A visual inspection of all sites indicated that the crown delineation consistently produced

crown layers that looked realistic when overlaying the imagery. We recognize, however, that it is extremely difficult to accurately delineate tree canopies in areas where crowns overlap. In some cases, a large tree crown may be falsely divided into small canopies or a cluster of shrubs may be grouped together into one crown (Rasmussen et al., 2011). It is important that the rate of falsely divided and falsely grouped crowns is balanced since excessive division of large trees into smaller leads to higher estimates of both crown density and aggregation. We evaluated the performance of

the classification and delineation methodology using field data from sites in Kenya (Appendix A). This showed that crowns smaller than ~2 m diameter were not reliably detected in the imagery. The validation analysis resulted in relatively strong agreement between estimated and field measured woody properties with R-squares of 0.69 (mean crown size), 0.82 (crown density), and 0.77 (woody cover) when crowns smaller than 2 m diameter were removed

from the field data set. We did, however, find that particularly large and spread-out crowns were subdivided, leading to underestimation of crown sizes and overestimation of crown density.

### 2.4 Environmental variables

The rainfall data were extracted from the Tropical Rainfall Measuring Mission (TRMM) 3B42 v7 product (0.25° x 0.25°) for the years 1998-2015 (Huffman et al., 2007). In addition to mean annual precipitation (MAP), we used rainfall seasonality represented by the coefficient of variation of mean monthly rainfalls. For soil data we used the sand content in the top soil layer (0-5cm) from the ISRIC/AfSIS 250 meter soil property maps of Africa (Hengl et al., 2015). To represent topography we used slope (%) derived from SRTM v2 (3 arc-seconds) elevation data (Farr et al., 2007). Fire frequency (fire events/year) was calculated using the MODIS MCD64A1 collection 5.1 burned area product (500m resolution) for the years 2001-2015 (Giglio et al., 2009). To avoid registering fires identified in adjacent months as separate fires, we counted fire events in consecutive months as a single fire. The extraction of raster values was based on nearest neighbor to the center point of each site in all cases except the TRMM data, for which we used bilinear interpolation due to its coarse resolution.

### 2.5 Statistical analysis of woody vegetation properties and the local environment

We derived four statistical properties of woody vegetation from each image: mean crown size (m²), density (crowns/ha), woody cover (%), and spatial aggregation of woody plants. Aggregation was calculated from the center points of the crown polygons. We used Ripley's K transformed to Besag's L-function to estimate aggregation at distances from 1 to 60 m (Besag, 1977; Ripley, 1977). Calculations were made using the spatstat R package with isotropic edge correction. The L-function was normalized by subtracting the distance so that 0 represents a random pattern and positive values indicate aggregation. For the analysis, we used the L-function at 20 m to represent aggregation as this distance is longer than the typical diameter of savanna trees and within length-scales of facilitative tree-tree effects. When analyzing crown sizes and aggregation, we excluded all sites with a crown density of 10 crowns/ha or less due to their low sample size for these metrics. We used boosted regression trees (BRT, in the dismo R package) to relate woody properties to the environmental variables. Its advantages include the ability to model non-linear relationships and to identify interactions between variables (Elith et al., 2008). When generating the BRTs, we used family = gaussian, tree complexity = 3, learning rate = 0.01, and bag fraction = 0.5 as model parameters. $R^2$, calculated through 10-fold cross-validation, was used for evaluating the strength of the relationships. The influence of individual predictors was estimated from their relative importance in the BRT models, and the directions of relationships were inferred from their trends in partial dependence plots.

The dataset includes several sites with PVPs, which often are treated as a special case because of their striking appearance (Figure 4). It is of interest to examine the environmental conditions associated with the occurrence of PVPs as well as those associated with aggregated woody populations in savannas without PVPs. We therefore separated sites with periodic vegetation from the rest and generated an additional set of models. The category with periodic vegetation contained 149 sites situated in Somalia, Senegal, Chad, Mali, Niger, Namibia, and Sudan. The

identification was based on visual inspection, and all sites with traits of periodic patterning (spotted, labyrinthine, gapped or banded) were put in the PVP category. We created one model for predicting aggregation among all sites, one for predicting aggregation among sites with no PVPs, and a third for predicting the occurrence PVPs. In the latter model, all PVP sites were given the value 1 and the rest 0, and the BRT family parameter was set to "bernoulli", appropriate for binomially distributed data.

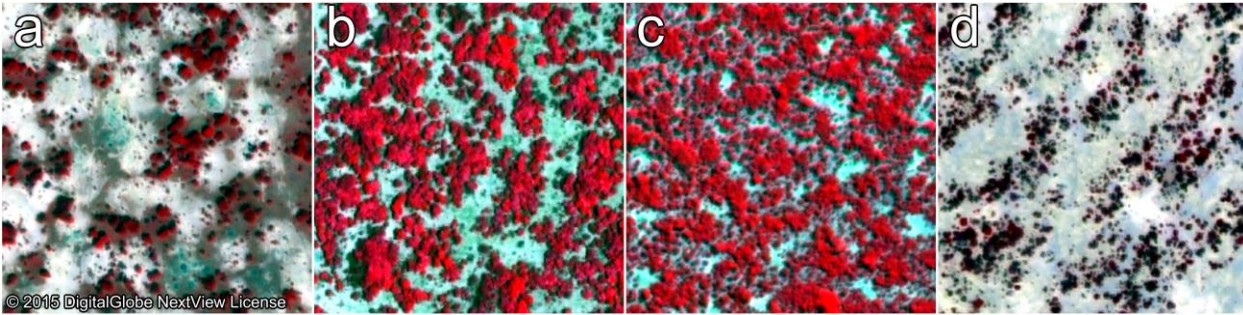

**Figure 4: False-color imagery of periodic vegetation patterns identified among the sites: (a) spotted pattern in Senegal, (b) labyrinthine pattern in Mali, (c) gapped pattern in Niger, (d) banded pattern in Somalia. Sites with PVPs were identified visually by the authors.**

## 3 Results

We started by calculating frequency distributions of the four woody properties divided into three rainfall categories (Figure 5). The more arid savannas (<400 mm/year) typically featured smaller crown sizes, lower crown density and woody cover, and higher levels of aggregation than sites in the wetter categories.

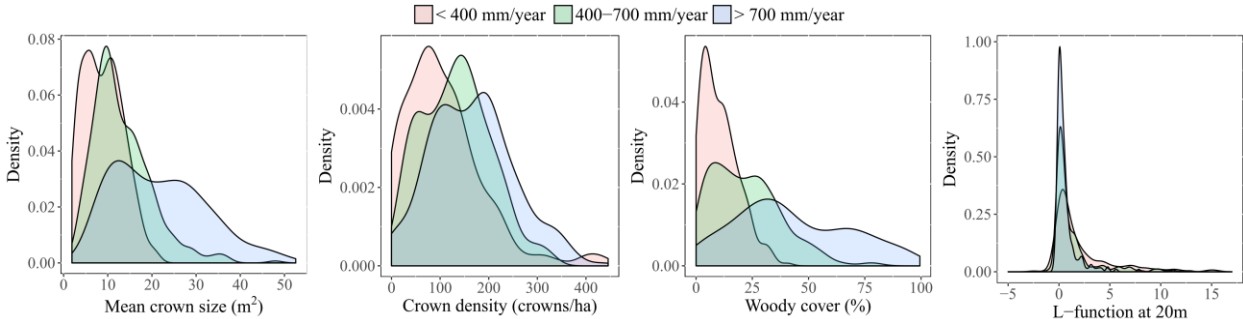

**Figure 5: Frequency distributions of mean crown size, crown density, woody cover and aggregation calculated for different MAP ranges.**

### 3.1 Mean crown size, density and woody cover

Boxplots with woody properties divided into MAP bins (Figure 6) show that woody cover and crown sizes increased more sharply with increasing rainfall than crown densities. Along the rainfall gradient from the driest (<200 mm/yr) to the wettest (1200-1400 mm/yr) end, mean estimates of crown size, crown density, and woody cover increased by 233 %, 73 %, and 491 % respectively. The BRT models for woody cover and mean crown size had high cross-validated

R² (0.73 and 0.68) and the same environmental factors that control woody cover also had a large influence over crown sizes (Table 1). In both cases, MAP had the largest relative influence followed by rain seasonality. While MAP had a clear positive influence on both woody cover and crown sizes, it was more difficult to interpret the influence of rain seasonality (Figure 7). Woody cover had a weak unimodal response to sand content, that was driven by the relationship between crown size and sand content (Figure 7). Fire frequency resulted in weak negative responses on all woody properties.

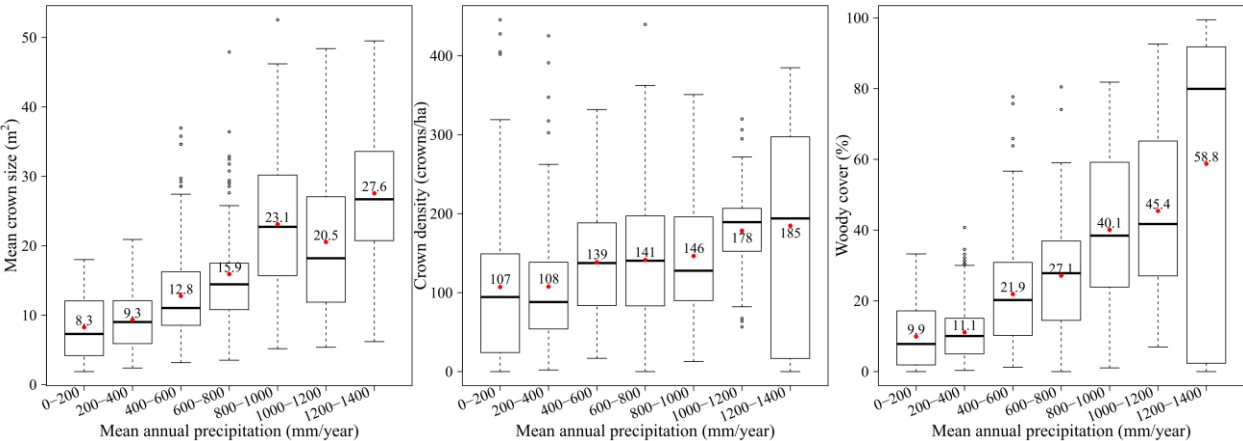

**Figure 6: Boxplots of estimates of crown size, crown density, and woody cover along a rainfall gradient. Red points denote the means. Between the driest (<200 mm/yr) and wettest (1200-1400 mm/yr) categories, mean estimates of crown size, crown density, and woody cover increased by 233 %, 73 %, and 491 % respectively.**

**Table 1: Relative influence of each environmental variable and the cross-validated R² from the BRT models when modeling woody cover, crown density, and mean crown size.**

| Variables | Mean Crown Size | Crown density | Woody cover |
|---|---|---|---|
| MAP | 45% | 33% | 47% |
| Rain seasonality | 21% | 37% | 23% |
| Sand content | 17% | 13% | 10% |
| Slope | 11% | 13% | 10% |
| Fire frequency | 6% | 4% | 11% |
| Cross-validated R² | 0.68 | 0.49 | 0.73 |

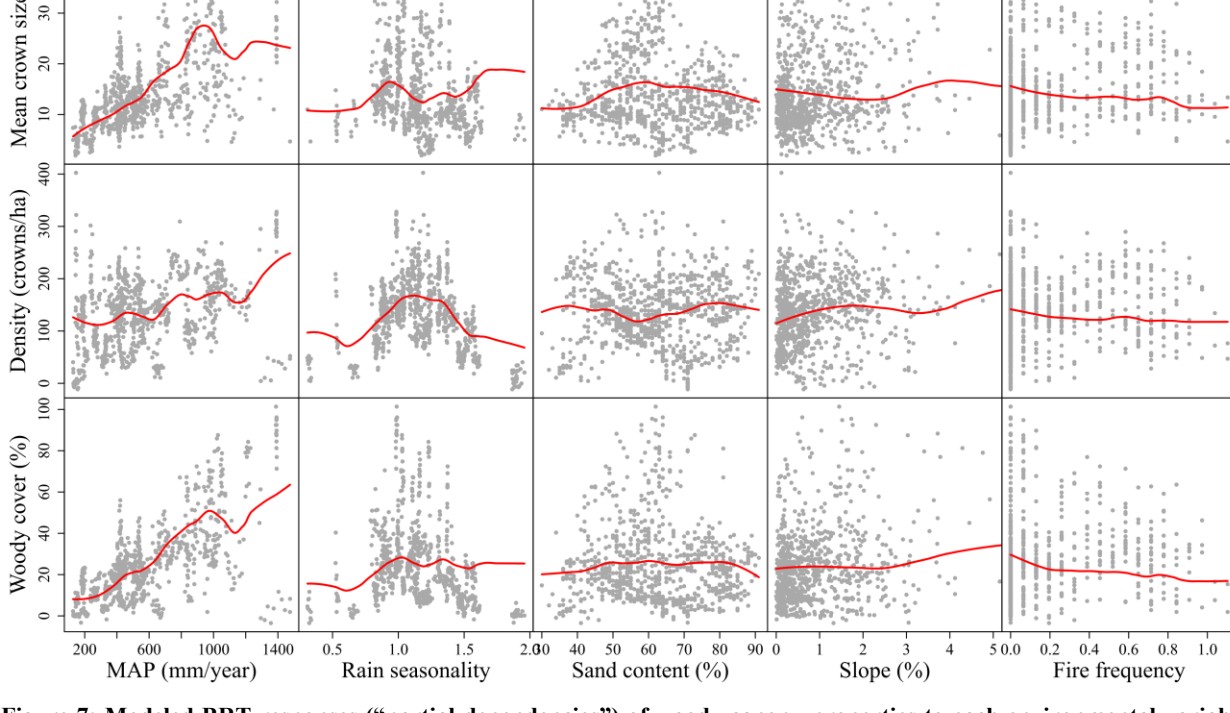

**Figure 7: Modeled BRT responses ("partial dependencies") of woody canopy properties to each environmental variable when accounting for the average effect of the other four variables. The red lines are smoothed representations of the responses, with fitted values (model predictions based on the original data) for each of the 876 sites shown as grey dots. The x-axis for the slope predictor was truncated at 5% to highlight the response in the bulk of the data.**

### 3.2 Woody plant aggregation

Our estimates of aggregation were based on the L-statistic (at 20 m) minus the distance, meaning positive values signal aggregated woody populations and negative values indicate dispersed populations (Figure 8). The large majority (82 %) of sites had positive values, indicating a rarity of dispersed woody populations in African savannas. There was little difference in the results for aggregation when sites with periodic patterns were included or not (Table 2). Higher levels of aggregation were generally associated with high seasonality, low MAP, fine-textured soils, and relatively flat terrain. These factors were also influential in determining the areas where periodic vegetation patterns occur. In fact, periodic patterns were absent in areas with MAP above 750mm, rain seasonality below 1.1, a sand content above 75%, and slopes steeper than 3.8%.

**Table 2: Relative influence of each environmental variable and the cross-validated R² from the BRT models when modeling woody aggregation (L-function at 20 m) and occurrence of PVPs. In the latter model, all sites with PVPs were given the value 1 and the rest the value 0.**

| Variables | Aggregation (all sites) | Aggregation (non-periodic sites) | Occurrence PVPs |
|---|---|---|---|

| | | | |
|---|---|---|---|
| MAP | 28% | 16% | 20% |
| Rain seasonality | 44% | 51% | 46% |
| Topsoil Sand | 14% | 16% | 21% |
| Slope | 14% | 17% | 1% |
| Fire frequency | 1% | 0% | 10% |
| Cross-validated R² | 0.31 | 0.29 | 0.83 |

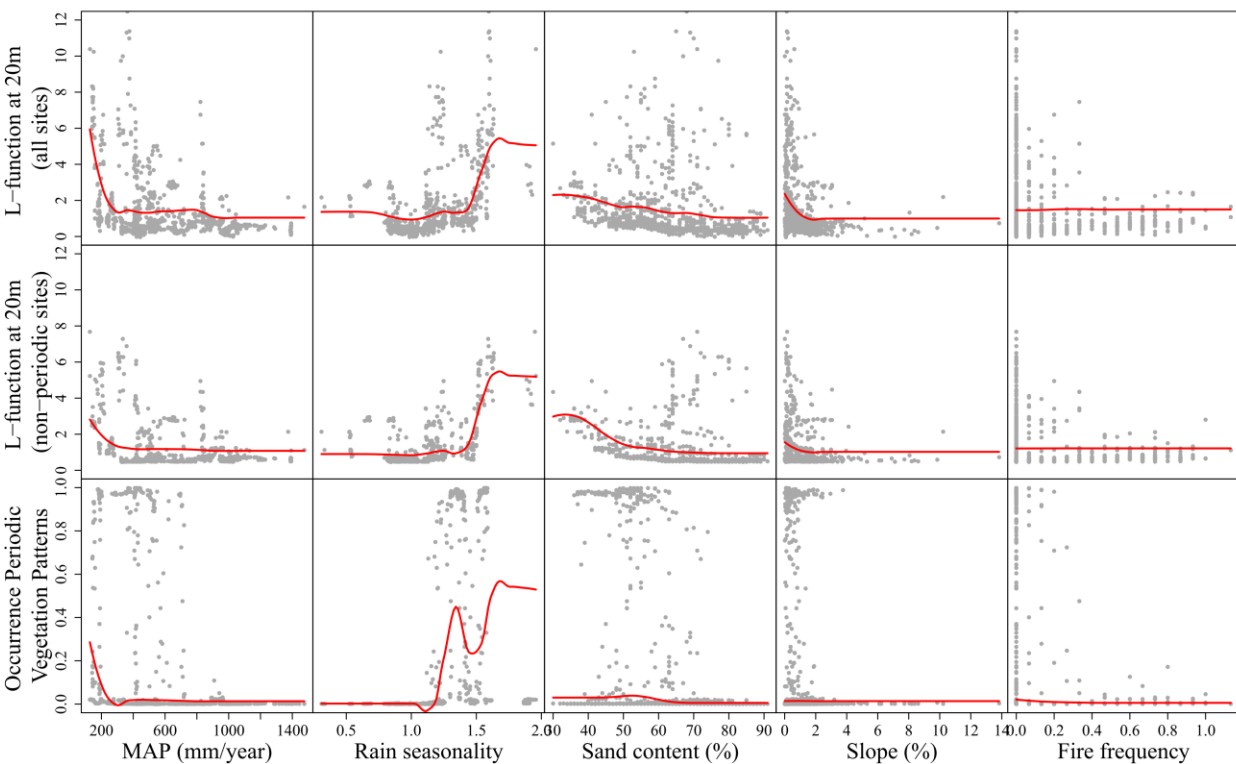

**Figure 8: Modeled BRT responses ("partial dependencies") for predictions of under what conditions PVPs occur (top), and woody aggregation (L-statistic at 20 m) for all sites not categorized as having periodic patterns (bottom). The response for each environmental variable accounts for the average effect of the other four variables. The red lines are smoothed representations of the responses overlaying the fitted values (model predictions based on the original data; grey dots).**

Additional insight was drawn from analyzing aggregation along distances and with the data categorized into PVPs and subdivisions based on MAP and soil texture (Figure 9). All categories were dispersed at short distances because each crown takes up space and there is bound to be a short distance between the center points of adjacent plants. Sites with PVPs had the highest levels of aggregation reaching a maximum at around 25 m (Figure 9). The combination

with wetter climes (≥600 mm MAP) on coarse-textured soils (≥60% sand) featured lower levels of aggregation than the other categories.

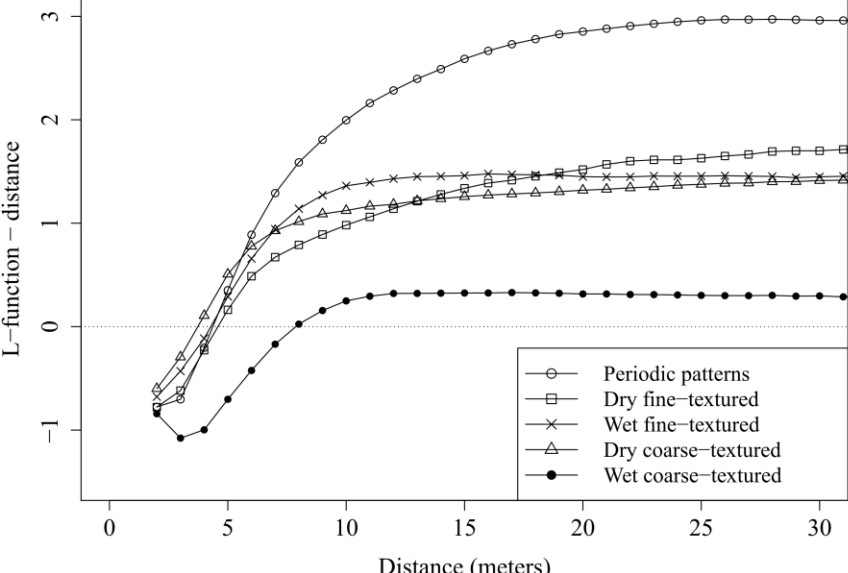

**Figure 9: Level of aggregation among tree crowns calculated using Ripley's K transformed to Besag's L-function. The figure shows the mean values of five categories: sites with periodic vegetation patterns, and four subdivisions based on mean annual precipitation and soil texture. Sites classified as having periodic patterns were not included in the latter subdivisions. Sites with MAP below 600 mm were categorized as dry whereas sites with a sand content below 60% were categorized as fine-textured.**

## 4 Discussion

### 4.1 Dividing woody cover into density and crown size components

Numerous authors have investigated how woody canopy cover varies across African savannas in response to variation in environmental variables (Good & Caylor, 2011; Sankaran et al., 2005; Staver et al., 2011). Given that tropical savannas cover about an eighth of Earth's land surface (Scholes & Archer, 1997) and contributes heavily to the global carbon cycle (Poulter et al., 2014), it is important to understand the makeup of these variations in terms of crown sizes and tree densities.By separating woody cover into mean crown size and density we were able to analyze whether they respond differently to environmental factors and how they combine to drive landscape-scale woody cover across the continent. Our results suggest that crown sizes respond more strongly to rainfall than crown density (Figure 6). This indicates that the commonly observed relationship of increasing woody cover with MAP in African savannas (e.g. Sankaran et al., 2005) is more a result of increasing size of trees than increasing tree density, at least in savannas with MAP < 700 mm. We also found a unimodal relationship between crown sizes and soil texture that was not present in the results for crown density (Figure 7). Soil properties have a considerable effect on the water cycle and a few studies have noticed that woody growth is suppressed on clayey soils in drylands (Lane et al., 1998; Sankaran et al., 2005; Williams et al., 1996). Recently, Fensham et al. (2015) showed that the effect is likely due to the higher wilting point on clays which limits the soil moisture available for plants to extract. A combination of low rainfall and fine-textured soils can lead to very low soil water potentials and impact the vegetation in a way reminiscent of even dryer conditions.

In our results, the relationship appears unimodal with suppression on both the clayey and the sandiest end. Woody growth is then controlled by available soil moisture which can be limited by either a high wilting point on clayey soils or low field capacity on sandy soils. Our results suggest these constraints affect the size of woody plants and not their abundance. Crown densities were most strongly influenced by rainfall seasonality and appears to have a unimodal response function (Figure 7). The sites with very low rainfall seasonality (<0.8) were all situated in the western part of East Africa (Serengeti, Masai Mara, and northern Uganda) in a region with bi-modal rainfall distributions and far lower seasonality that further east. Many of these sites had low woody densities and cover but likely for other reasons than rainfall seasonality. Elephant densities are thought to be a key driver of woody cover in the Mara-Serengeti ecosystem (Morrison et al., 2016). Browsing, especially by elephants, has a great impact on woody structure (Sankaran et al., 2013) and is a key factor we did not capture in this analysis. If we focus on sites with rainfall seasonality above 0.8, there is a more linear relationship with lower crown density and cover in areas with high rainfall seasonality which could be associated with the long periods of high water stress in more seasonal systems. Lehmann et al. (2014) found that high rainfall seasonality can constrain canopy closure and is an important predictor for the presence of savanna. Overall, the estimated woody properties were more strongly influenced by rainfall amounts and seasonality than by soil, slope, and fire. Fire frequency had a weak negative association with both woody cover, crown sizes, and densities. Fire has, however, an interactive relationship with vegetation structure (Archibald et al., 2009) and this analysis cannot separate the effect of fire on vegetation from impacts of vegetation structure on the fire regime.

**4.2 Woody plant aggregation and the occurrence of periodic vegetation patterns**

In accordance with previous research (Deblauwe et al., 2008), we found that the formation of highly aggregated PVPs is associated with specific environmental conditions. Periodic patterns are most likely to occur in areas with high rainfall seasonality, low mean annual rainfall, on fine-textured soils, and on flat or gently sloping terrain (Figure 8). These are factors that influence ecohydrological processes such as the propensity to form overland flows during rainfall events (Valentin et al., 1999). The results are in agreement with a global study on the biogeography of PVPs by Deblauwe et al. (2008) who found similar effects in regions with strong seasonal variation in temperature and more constant rainfall (Australia and Mexico) and in regions with distinct rainfall seasonality but more constant temperatures (Africa). Our analysis further shows that the same factors that contribute to PVP emergence are associated with higher levels of aggregation among woody plants elsewhere in African savannas. PVPs thus appear under conditions that naturally favor local facilitation and patchiness. However, the vegetation at many sites with these conditions do not exhibit highly organized periodic patterns which could be related to soil properties other than texture. The dominant process in the formation of PVPs is a significant overland flow from bare to vegetated patches which requires near impervious soils. This property is typically associated with shallow soil depths, physical crusts, or hardpans (Leprun, 1999; McDonald et al., 2009), and is not strongly dependent on soil texture.

Previous literature have linked local aggregation and patchiness in savannas to fire frequency (Veldhuis, Rozen-Rechels, et al., 2016), seed dispersal (Pueyo et al., 2008), runoff-erosion processes (Ludwig et al., 2005), and short-range facilitation through modified microclimate close to nurse plants (Holmgren & Scheffer, 2010). With increasing

abiotic stress, we expect stronger tree-tree facilitation in accordance with the stress gradient hypothesis (He et al., 2013). In our analysis, the most influential predictor for modeling aggregation was rainfall seasonality (Table 2), a factor that could influence plant dynamics in more than one way. The pronounced dry season associated with highly seasonal systems exerts a strong abiotic pressure, especially on juvenile trees with less developed root systems. Juvenile survival through the dry season is likely higher in the shelter of nearby trees. Over time, a bias in survival rates may lead to higher aggregation among adult trees. Once the wet season arrives, it often comes in heavy downpours which can quickly saturate the top soil leading to overland flows. This leads to both redistribution of water resources to woody patches with higher infiltration rates, and redistribution of litter and soil resources (Ludwig et al., 2005). The more concentrated rains may also alleviate competition for water during the growing season leading to facilitation being the dominant force in highly seasonal drylands. There was also a clear relationship between fine-textured soils and higher aggregation. Fine-textured soils increase runoff through lower infiltration rates, and may also amplify stress during the dry season through their higher wilting point. Sites with the combination of coarse-textured soils ($\geq$60% sand) and  wetter climes ($\geq$600 mm MAP) stood out in the analysis by being far less aggregated (Figure 9). This points to the interactive effects of these variables. We found no link between fire frequency and aggregation and a weak relationship with slope favoring aggregation on flat or gently sloping terrain. This relationship can also be explained in terms of overland flows. Steeper slopes tend to create drainage rills leading the water downhill which break up the local patch-interpatch redistribution of resources (Saco & Moreno-de las Heras, 2013).

**5 Conclusions**

Using high spatial resolution imagery, a flexible classification framework, and a crown delineation methodology, we estimated several key woody vegetation properties in African savannas and analyzed how these vary with local environmental conditions. We find that woody cover, crown sizes, and woody plant densities are more strongly influenced by rainfall amounts and seasonality than by soil texture, slope and fire frequency. Of specific interest is that mean crown size responded more strongly to mean annual rainfall than plant densities, indicating that the commonly observed relationships between woody cover and rainfall (e.g. Sankaran et al., 2005) is more a result of increasing crown sizes than changes in crown density. Larger crown sizes were associated with mid-textured soils and appeared suppressed on both clays and very sandy soils. The level of aggregation among woody plants was most strongly related to rainfall seasonality, as was the occurrence of PVPs. Similar processes that influence patchiness in savannas also contribute to the formation of PVPs, with impermeable soil conditions being the possible difference between a patchy savanna landscape and highly organized periodic vegetation.

**Acknowledgements**

The satellite data were provided through a NASA agreement and under NextView license. We thank Jamie Nickeson, Njoki Kahiu, Sujan Parajuli, and Dinesh Shrestha for assistance with data retrieval, field work, and image classification. The project was funded by the National Science Foundation (Coupled Natural-Human Systems Program) and the NASA Terrestrial Ecology Program. CRA was also supported by a Graduate Research Fellowship through the Geospatial Sciences Center of Excellence at South Dakota State University.

**Appendix A: Validation of Estimated Woody Vegetation Properties using Field Data from Kenya**

This appendix describes a validation analysis of estimated mean crown size, crown density, and woody cover using field data collected in southern Kenya during September-October 2015. Plots were established in five protected areas: Tsavo West NP, Tsavo East NP, Amboseli NP, Ol Pejeta wildlife conservancy, and Il Ngwesi group ranch (Figure

A1). In total, we established 28 plots with at least four plots in each protected area. The size of plots varied with the density of trees and shrubs, ranging from 350m² to 8000m² with a median at 1450m² (38x38m). The position of plot corners were determined with a GPS and the positions of trees and shrubs within each plot were measured with a laser rangefinder from the plot corners. Using measuring tape, we determined the diameter of crowns along the longest axis and on the perpendicular. From these two measurements, we later calculated crown sizes assuming elliptic crown

shapes. We acquired the best available high resolution imagery covering the sites from 2012 or later. In some cases, this resulted in imagery of lower quality (few green leaves on the trees) than the imagery used in the continental analysis.

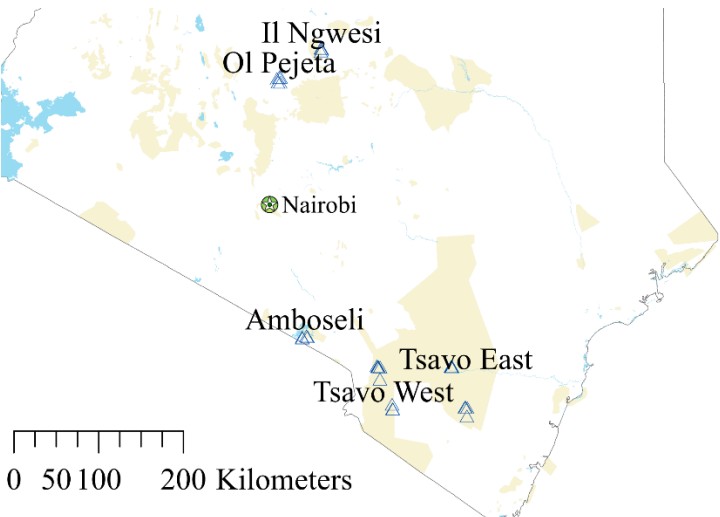

**Figure A1: Map of the five protected areas in southern Kenya where field work was conducted. The positions of individual**
**plots are marked with blue triangles.**

Our analysis of detection ratios (Figure A2) indicated a detection threshold of ~ 2 m below which smaller trees and shrubs were not reliably detected, while most individuals with crown diameter > 3 m were detected. The detection ratios were likely negatively influenced by the sometimes low quality of the imagery and the time difference between image acquisition and field work (often 2-3 years).

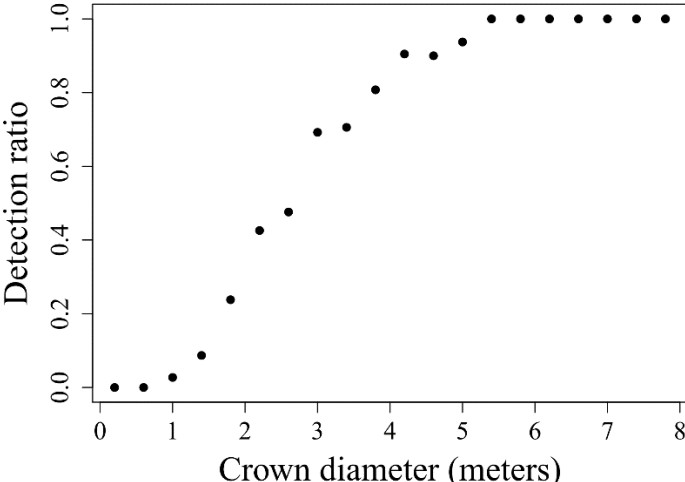

**Figure A2: Detection ratios of woody plants in classified imagery at field work sites. The values were calculated as mean detection ratios for trees divided into bins of width 40 cm.**

When calculating the relationship between estimated and field measured woody properties (Figure A3), we excluded all field measured trees and shrubs with a diameter less than the 2 m detection threshold. Estimates of the woody

properties then fall relatively close to the one-to-one line. The four sites in Amboseli were dominated by large umbrella thorn acacias (*Vachellia tortilis*) with particularly large and spread out crowns (Figure A4). The spread-out architecture of these crowns make them appear as several distinct crowns from above, and the delineation algorithm did not identify them as single trees. The large majority of our sites in the continental analysis do not contain this type of trees which are relatively rare across all of African savannas. Since they dominated all four sites in Amboseli, we

determined they were overrepresented in the field data set and therefore chose to exclude the Amboseli sites when calculating R² for mean crown size and crown density. We also excluded one site in Ol Pejeta (OLP3) where the smaller trees lacked green leaves in the imagery and could not be detected.

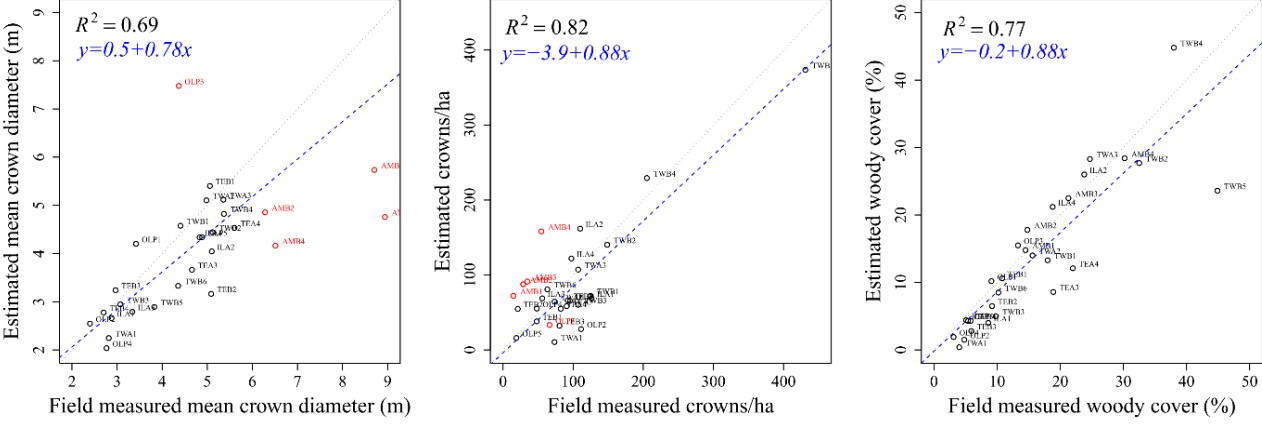

415 **Figure A3: Validation of estimated mean crown size, crown density, and woody cover. The Amboseli sites and one site in Ol Pejeta were excluded when calculating R² for mean crown size and crown density. These sites are shown in red color.**

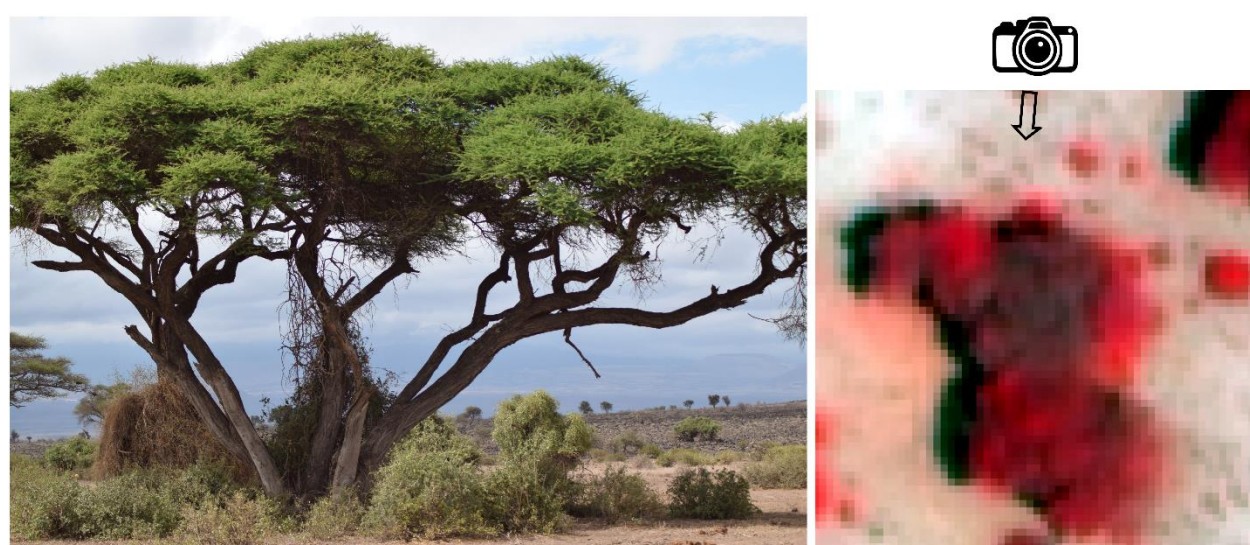

**Figure A4. Vachellia tortilis at a field work site in Amboseli NP, Kenya. The left image shows two trees with overlapping crowns, with the second being further back on the left. The right image shows the same trees in false color satellite imagery.**
420 **The spread out architecture of the canopy make them appear as several distinct crowns. The camera symbol roughly indicates the position from which the ground photo was taken.**

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
