# Peer review of "Patterns in Woody Vegetation Structure across African Savannas"

_Biogeosciences, 2017_

## Short Comment (SC1) · 25 Jan 2017

The authors state: "Because of uncertainty in the accuracy of the woody properties derived from the delineated crowns, we do not focus on absolute numbers but on how they vary across environmental gradients."

The study assumes that errors in the woody proportion estimation are either negligible or are the same across the environmental gradients and at the different African sites. What evidence is there for this ?

Tree crown morphology (roundness etc.) and remote sensing signal contributions (i.e., soil, understory and shadow) are unlikely to be the same across the African sites/gradients and so errors in the empirical crown delineation method are likely to be variable. Moreover, the tree crown delineation is applied to different satellite data

(WorldView, GeoEye, Quickbird).

Without a quantitative accuracy assessment of the tree crown delineation across the gradients/sites and among the data used this interesting study is incomplete and potentially flawed - the results may be controlled by crown delineation differences.

---

## Short Comment (SC2) · 1 Feb 2017

As part of a course in Critical Thinking at The University of Edinburgh, myself and colleagues chose to review your paper, each taking sections to critique. We would like to thank you for this opportunity to comment and we hope that our input will be beneficial to your work.

The following comments relate to the Discussion section of the paper.

1. In Discussion, the relationship between woody cover, crown size and crown density is introduced (Line 225), yet seems not to be mentioned in any section before – perhaps it would be fitting to highlight this relationship in an earlier section.

2. In line 228, the term 'woody density' is used, causing the reader some confusion, as

previously the distinction was made between woody cover and woody density. Is this new term intentional?

3. We had some queries surrounding the focus on sites with rainfall seasonality above 0.8 (Line 245) - what is the justification for this focus? Is this where a sensitivity lies in terms of woody density response, and is this a causative relationship? Some more detail here would benefit the discussion.

4. On line 243, 'previous literature' is mentioned but no references are given immediately. Although individual associations are then explained and cited, the opening sentence omitting these references could be improved by restructuring this paragraph.

5. The use of the word 'So' to open line 268 does engage the reader, but seems inconsistent with the more formal writing style in the rest of the paper.

General comments on the study Ecosystem services is mentioned near the beginning of the paper but is not revisited in discussion – how relevant are ecosystem services to this study? 'Processes' seem like a more accurate description of relationships mentioned. Although the effects of primary variables were investigated, no interaction effects investigated. Are there any coupling effects of say, rainfall and slope on woody density?

---

## Short Comment (SC3) · 1 Feb 2017

As part of a university course in critical thinking, we chose to read this paper as a group and submit comments for each section of it individually. The comments below will address the Methods section.

The last paragraph of the introduction seems to go into rather a lot of detail, and some of the things discussed in it could perhaps have been kept for the methods section. The 'initial unsupervised classification with manual assignment into woody, herbaceous, and bare cover classes' is rather vague – it is not clear who was carrying out the classification or in what way they were unsupervised.

It would be useful to include the number of sites that were initially considered for inclusion in the study and the number that were eliminated, perhaps before stating the final

number of sites, to make it clearer that the value of 876 is after all sites not suitable for inclusion have been eliminated.

Some parts of the methods section lack justification for the decisions that were taken. For example, in line 132 the authors state that segments were not merged if the resulting crown size has a diameter greater than 40m, but there is no justification for why this particular value has been chosen, making it seem as if it is at random. In line 133, the authors state that they experimented with different methods before settling on the ones they actually used, but do not explain why they didn't use the previous methods; without this justification, mentioning that alternative methods were decided against seems to provide no insight.

It would be interesting to know how long the image classification process took, since the authors indicate that it was the most time-consuming part of the analysis – without knowing how long the analysis took in total this piece of information doesn't tell the reader very much, although it is interesting to know.

An estimation of the accuracy of the methods would be useful, especially concerning the delineation of tree crowns. It is possible there might be a bias in the estimation of crown size across environmental gradients, meaning that the rate of falsely divided and falsely grouped crowns may be different at different values of environmental factors. If this is the case even comparing across environmental gradients could be inaccurate. In line 150 the authors state that the generated crown layers 'look realistic from a visual inspection across all landscape types and different tree densities', but this is a very qualitative way of assessing the accuracy of the methods; a quantitative value of accuracy would be better.

---

## Short Comment (SC4) · 1 Feb 2017

The following comments are the result of a group discussion as part of a university course on Critical thinking.

Comments on Abstract

The abstract gives a good overview of the scope of the paper. However, the same amount of information could probably be conveyed with fewer words. Additionally, the description of the results is particularly detailed but not quantitative enough. Finally, the abstract would benefit from concluding sentence.

Comments on Results section

You mentioned that there was a "strong relationship" between woody cover and mean

crown size (line 197). However, you did not state any threshold in regards to that value or how you arrived at it. Similarly, you referred to the "large majority" of sites (line 206) without specifying exactly how many. Another suggestion that we have about the results section, is to consider the interaction effect between the variables in the analysis. Generally, it is not entirely clear whether you answered your research question and how you can quantify the success of this research project. It is also more convenient to have the graphs and figures within the text rather than in an appendix but we assumed this is because the paper is still undergoing review.

---

## Short Comment (SC5) · 2 Feb 2017

During a group discussion of this paper several points were raised. I shall discuss the issues form the introduction and conclusion sections.

Introduction

1. On line 35 the authors state "climate, both rainfall patterns and temperatures, could change in many parts of Africa" and then the reference. It would benefit the reader if some of these changes, as they are relevant to your research, were stated and the effect that these changes could have on vegetation.

2. The use of words like 'these' and 'those' should be avoided to remove any ambiguity about the subject of the statement.

3. The end sentence in the second introduction paragraph (beginning line 51) could be improved to reduce vagueness; how do we learn about the impacts of the underlying ecosystem processes?

4. Aggregation is discussed a lot but there is relatively little introduction to it. The relationship between woody plants and aggregation should be given some context.

5. A diagram of the spotted, labyrinthine, or gapped patterns of PVPs would be useful for readers less familiar with PVPs.

6. The last paragraph of the introduction seems out of place, and would be better suited for the methods section. A smaller summary of your work would be appropriate for the introduction, and Figure 1 would definitely be better placed in the method section.

7. It would be better to end the introduction with a research question, or the aim of your experiment. A lack of clear hypotheses made it hard to read the results and judge whether the experiment was successful or not. A clear research question helps the reader know what you are trying to achieve.

Conclusion

1. The conclusion was more of a summary of your results, rather than a rounding up of the issue explored. Ideas for future work could be given, and the importance of the work restated.

2. The phrase "possible difference maker" is a clumsy end to the paper.

---

## Referee Comment (RC1) · P. Mograbi (Referee) · 16 Feb 2017

General comments

This paper explores abiotic drivers of woody vegetation density and crown size across African savannas. Here, woody cover is seen as a function of tree density and crown size. The authors find that increasing woody cover results in the literature are attributable to increasing crown size rather than increasing tree density, which is a controversial topic and would contribute significantly to terrestrial carbon pool debates, as well as the bush encroachment sphere. The authors also assess abiotic drivers of regular vegetation patterns and the possible ecohydrological processes behind the patterns.

This was an interesting paper to review – interesting content and thorough methods

section. The content is appropriate to the scope of BG and does contribute to land-vegetation interactions in general.

The paper was mostly well-written and I noticed almost no errors in text. The discussion was particularly well-written, with interesting links to other studies and well-thought out exploration of explanations for the relationships between woody vegetation properties and the abiotic template. The data processing and methods section were detailed and clear, with parameters, limitations and steps clearly presented. Most of the technical considerations/challenges/limitations that occurred to me after I read the abstract and got a first broad overview of the methods were explicitly and thoroughly dealt with in the methods section. The authors are well aware that the crown size/density metrics are probably not that accurate, but that there is still merit in assuming that the error has been propagated consistently over space and it is in the relative differences that insight into vegetation patterns are shown. That said, I have a few points:

1) For me, the hinge-point of this study's methods are that tree crown center points are derived from relative NDVI differences. While this might be valid (and from an eye-ball of Figure 3 it seems to work), there are no references discussing this method. I would suggest this method be backed up by previous references. I would also like to know what the limitations of this method are. I would also like to see some form of validation stats (e.g. Kappa) for the accuracy of the woody cover/forest mask, crown size, crown density outputs. Perhaps some test sites could be manually evaluated and compared with the semi-automated approach. You mention "uncertainty in the accuracy" of your metrics on ln 151 so perhaps the authors have already performed an error test and haven't reported it? It would be interesting for readers to know how well these methods performed (and it would increase your citations!)

2) The counterpoint to well-written discussion, is the introduction is not the same quality and reads like a rough draft. The introduction lacks the key "introduction linkage" points made both in the abstract and the discussion. The introduction and discussion should book-end the findings, and the introduction was inadequate in this regard. While the

content for the motivation and aims for the study were available if one was looking for it, they were not presented in a clear flow and it felt weak. There were also several lines that would be better suited in the methods section. I have made suggestions on how the introduction could be improved below in the specific comments.

3) I intuitively felt an important part of this study was mentioned in the discussion for the first time. Ln229-230: The results of your study suggest that increasing woody cover trends from multiple previous research articles are related to increasing crown size, rather than increasing density. This is huge and forms the central finding but is only mentioned once! There are important implications for global carbon cycles (see Poulter et al. 2014 Nature and Liu et al. 2015 Nature Climate Change), bush encroachment etc. This would be a finding that other scientists would explore further. You need to develop this theme. I want to know more!

4) PVPs: This aspect of your study is mentioned briefly in the beginning, forms a large chunk of your results and more than half of your discussion. This leaves the manuscript unbalanced and the reader is left wondering why PVPs are so important and why it was decided to explore it so heavily. If the focus is on PVPs, that needs to be reflected in the abstract (it isn't mentioned once until the end) and introduction (it is mentioned as a phenomenon but no why they are worth exploring or what the question is about them. While I am no specialist in PVPs, I would also suggest that no lit review section of PVPs is complete without a mention of Max Rietkerk's work, particularly Rietkerk & van de Koppel 2008 Trends in Ecology and Evolution. I was also missing mention of Bromley et al. 1997 Journal of Hydrology which specially mentions West African PVP's and 'tiger bands'.

Specific comments

1. Title: The title has the word "Savannas" in it. Yet, later on the authors mention 'drylands' (ln 40) which contains large areas not typically counted as savannas. In Figure 2 the vegetation area of interest is labelled 'rangelands', as well as in ln 87.

Why use Ellis & Ramankutty's anthropogenic biome for an abiotic-vegetation study on savannas when you could use a climatic-disturbance based biome which defines savannas? This study does not consider the human component. Whichever term the authors choose require clarification and should be used consistently. Why not provide a map of the savanna extent? For example, the 'rangeland' areas in Morocco, Algeria and Libya are traditionally not considered savannas. The authors could use the extent used by Sankaran et al. 2005 Nature as it is widely accepted. It could even be interesting to see the relative differences in abiotic influences on your sites divided into the "stable" and "unstable" savanna categories, if they agree with Sankaran et al. Just a thought. The other issue with the title is the word "structure" when your metrics measure woody cover and tree density. My understanding is that 'structure' implies height metrics or SCD's.

2. Abstract: PVPs not mentioned until ln 28. They need to be introduced earlier if they are the focus of the study.

3. Ln 12-15 Very concise and clear summary of your introduction and aim in these abstract sentences. This idea also needs to be explicitly stated upfront in the introduction and well referenced. I probably lost the impact of this point in the introduction because of poor flow and structure.

4. ln33-34 "While humans often play a dominant role in many systems..." I did not understand the point of this statement and it feels out of place here. Either remove it or expand on it.

5. ln 38 "...future stability and productivity..." 'stability is a loaded term in savanna literature. Perhaps rephrase this. This idea would form a nice link to bring up again in your conclusion to tie your manuscript together.

6. ln 44-45 Great to bring up fire's influence. Recent work by Smit et al. 2016 Journal of Applied Ecology show that SCD's are affected by high intensity fires, including tall tree (large canopy size?) loss. I understand that fire intensity can't be ascertained with

MODIS data, but it does need to be mentioned that intensity plays a role.

7. ln 40-50 This may be a personal style preference, but worth a mention (word limit permitting). The first half of the paragraph lists abiotic driver influences on woody veg properties, the 2nd half specifies how these drivers can influence the specific metrics of the study (individual: crown size; population: crown density, woody cover) and provide an example of how the same woody cover can have different ecosystem functioning. This is a natural flow, but I wanted a bit more on both topics. Could these two sections be expanded to their own paragraphs?

8. ln 64-65. Please reference mention of vegetation bands. Rasmussen? Bromley?

9. ln 69-70. Both studies the authors cite for "African savannas" are from W Africa. Could other African studies be included?

10. ln +-73-81 This paragraph seems more suited to the methods section. Perhaps you could reduce these details to a sentence or two, linking the methodological processes to the general aim, rather than mention details here and then details again in the very next paragraph? Figure 1 should also only be mentioned in the methods.

11.ln 80 The PVPs identified in the study sites, were those sites derived from the literature or were they found by the authors. Please mention this. If the latter, it would be nice for the reader to have image examples of the different kinds of PVPs. Are they very easy to spot?

12. ln 90. Does this mean spring in the northern and southern hemispheres? Could you be more specific?

13. ln 91-92 It's not necessary to mention that another on-going study influenced this one's parameters unless some of the data from that study are included in this paper. Perhaps leave this out.

14. Methods: The sections on preprocessing and classification were thorough. Thanks!

15. ln 112-115 This section isn't really necessary for the article, although I do understand the feeling of wanting the time and monumental effort taken for analyses to be recognised by the readers!

16. ln 132 Is there a reason for the 40 m limitation to crown size?

17. lns 143-154 This is a well-needed section and I like that the limitations are mentioned. However, it needs bolstering with supporting literature. A quick google search has shown that crown delineation techniques with multispectral, high resolution satellite data exist and it would useful to see a comparison of the trade-offs to back up the method you have used. This ties in with my request to see support for the NDVI crown centre identification method. Accuracy statistics would be a useful addition here.

18. ln 172 Was a 20 m cut-off used for Ripley's L because that is where the sill occurs on all the curves in Figure 5?

19. ln 192 Mentions Figure 4. Figure 3 was never mentioned. Please include it where relevant.

20. Results: The subheadings seem strange. You have one sentence on vegetation characteristic differences followed by a subheading "3.1. Mean crown size, density and woody cover". Surely the previous paragraph (of one sentence) fits into this subheading? Or was the subheading meant to be related only to BRT results?

21. ln 194 It was not clear to me from Figure 4 that arid sites had higher levels of aggregation. Perhaps because the colours did not come out well in that panel?

22. ln 197 "Woody cover and mean crown size both had strong relationships with the local environment..." What factors in the local environment? Could you be more specific?

23. ln209-211 Nice findings. The sentence that starts " These are factors that influence ecohydrological processes...."at the end of the paragraph is better suited to the discussion section and needs to be referenced.

24. ln 219. "...aggregation reaching a minimum at around 25 meters." Consistency with meters/m. This sentence also needs a figure reference at the end. Figure 7?

25. ln 219-220. This is a discussion point.

26. "Heading 4.1. Dividing woody cover into density and crown size components" as well as ln 226-228 are concepts that should be addressed in the introduction. This is a key part of what makes this study novel as most research deals with woody cover without addressing density/crown size differences. These lines are the coherent aim and motivation I was missing in the introduction.

27. ln 229-231 Great finding! Make a meal of it. The authors need to discuss this vs. bush encroachment findings in the literature.

28. ln 243-245 Low woody cover unrelated to rainfall seasonality. This section needs mention of the large role of disturbance agents in "unstable savannas" (sensu Sankaran et al. 2005). The authors do mention elephant impacts in a sentence, but this needs more unpacking and forms part of the caveats to this study's results as biotic disturbance was not included. Together with acknowledging effects of fire intensity on SCDs.

29. ln 254 "In accordance with previous literature..." There are no references at the end of this sentence. Which literature?

30. ln 270 "...and short-range facilitation through modified microclimate close to nursery plants" needs a reference.

Technical corrections

1. ln20-30 Be careful of the change in tenses. Generally, methods and results should be reported in the past tense.

2. Journal editor preference, but Figure mentions should normally be in parentheses, rather than mentioned in the sentence.

Eg. "Frequency distributions of the four woody properties, separated into three rainfall

categories, are shown in Figure 4." To "The more arid savannas (<400 mm/year) typically feature smaller crown sizes, lower crown density and woody cover, and higher levels of aggregation than sites in the wetter categories (Figure 4)."

3. ln 118 Insert spaces between "240x240" and shouldn't 'meter' be 'm'. Be consistent throughout the manuscript. Either change previous mention of 'meter' to 'm' or vice versa.

4. ln 124 'ID' or 'point' rather than 'id'

5. ln 241. This is the only occasion a discussion sentence refers to a results Figure. Either include more links to the results where appropriate, or remove this one. Consistency. e.g. ln 228-229 could also use a figure reference?
* * *

---

## Author Comment (AC1) · 16 Feb 2017

We (the authors) are happy with the comments and the review on the paper this far. They will certainly help us update and clarify the manuscript to make it a better paper. One point that has been raised is the lack of validation and we have made a validation analysis that we upload with this comment. We plan to add it to the paper as an appendix. Responses to comments and the reviews will be posted later. Thank you!

Please also note the supplement to this comment:
http://www.biogeosciences-discuss.net/bg-2017-7/bg-2017-7-AC1-supplement.pdf

[Figure]

**Supplement:**

**Appendix A**

This chapter describes a validation analysis of estimated mean crown size, crown density, and woody cover using field data collected in southern Kenya during September-October 2015. Plots were established in five protected areas: Tsavo West NP, Tsavo East NP, Amboseli NP, Ol Pejeta wildlife conservancy, and Il Ngwesi group ranch (Figure A1). In total, we established 28 plots with at least four plots in each protected area. The size of plots varied with the density of trees and shrubs, ranging from 350m² to 8000m² with a median at 1450m² (38x38m). The position of plot corners were determined with a GPS and the position of trees and shrubs within each plot were measured with a laser rangefinder from the plot corners. Using measuring tape, we determined the diameter of crowns along the longest axis and on the perpendicular. From these two measurements, we later calculated crown sizes assuming elliptic crown shapes. We acquired the best available high resolution imagery covering the sites from 2012 or later. In some cases, this resulted in imagery of lower quality (few green leaves on the trees) than the imagery used in the continental analysis.

[Figure]

**Figure A1: Map of the five protected areas in southern Kenya where field work was conducted. The positions of individual plots are marked with blue triangles.**

Our analysis of detection ratios (Figure A2) indicates a detection threshold of ~ 2 m below which smaller trees and shrubs were not reliably detected, while most individuals with crown diameter > 3 m were detected. The detection ratios were likely negatively influenced by the sometimes low quality of the imagery and the time difference between image acquisition and field work (often 2-3 years).

[Figure]

20

**Figure A2: Detection ratios of woody plants in classified imagery at field work sites. The values were calculated as mean detection ratios for trees divided into bins with width 40cm.**

Figure A3 shows validation results for mean crown sizes, crown density, and woody cover. Here, we excluded all field measured trees and shrubs with a diameter less than the 2 m detection threshold. Estimates of the woody

25     properties then fall relatively close to the one-to-one line. The four sites in Amboseli were dominated by large umbrella thorn acacias (*Vachellia tortilis*) with heterogeneous crowns (Figure A4). The spread-out architecture of these tree crowns make them appear as several distinct crowns from above and the delineation algorithm did not identify them as single trees. While these trees are not rare, they were overrepresented in our field data set and we therefore chose to exclude the Amboseli sites when calculating R² for mean crown size and crown density. We also excluded one site

30     in Ol Pejeta (OLP3) where the smaller trees lacked green leaves in the imagery and could not be detected.

[Figure]

**Figure A3: Validation of estimated mean crown size, crown density, and woody cover. The Amboseli sites and one site in Ol Pejeta were excluded when calculating R² for mean crown size and crown density. These sites are shown in red color.**

35

[Figure]

**Figure A4. Vachellia tortilis at a field work site in Amboseli NP, Kenya. The left image shows two trees with overlapping crowns, with the second being further back on the left. The right image shows the same trees in false color satellite imagery. The branched-out architecture of the canopy make them appear as several distinct crowns. The camera symbol roughly indicates the position from which the ground photo was taken.**

40

---

## Author Comment (AC2) · 8 Apr 2017

We thank the reviewer for detailed and constructive comments on the manuscript. We are glad to get this kind of input. Below are responses to the comments.

General comments

1) For me, the hinge-point of this study's methods are that tree crown center points are derived from relative NDVI differences. While this might be valid (and from an eye-ball of Figure 3 it seems to work), there are no references discussing this method. I would suggest this method be backed up by previous references. I would also like to know what the limitations of this method are. I would also like to see some form of validation stats (e.g. Kappa) for the accuracy of the woody cover/forest mask, crown size, crown density outputs. Perhaps some test sites could be manually evaluated and compared

with the semi-automated approach. You mention "uncertainty in the accuracy" of your metrics on ln 151 so perhaps the authors have already performed an error test and haven't reported it? It would be interesting for readers to know how well these methods performed (and it would increase your citations!)

Response: Previous studies have used relative brightness (Bunting & Lucas, 2006) or vegetation indices e.g. NDVI (Karlson et al. 2014) to identify and grow crown segments. We referenced these studies without explaining in detail how they did it, and will expand on the description of these previous methods to better explain how they derive seeds for the crown segments. In terms of limitations and accuracy assessment, the added appendix should hopefully provide information on this. It contains results for a validation assessment using field site data from Kenya.

2) The counterpoint to well-written discussion, is the introduction is not the same quality and reads like a rough draft. The introduction lacks the key "introduction linkage" points made both in the abstract and the discussion. The introduction and discussion should book-end the findings, and the introduction was inadequate in this regard. While the content for the motivation and aims for the study were available if one was looking for it, they were not presented in a clear flow and it felt weak. There were also several lines that would be better suited in the methods section. I have made suggestions on how the introduction could be improved below in the specific comments.

Response: We will revise and expand the Introduction in response to this comment to improve the flow of the introduction and its link to the discussion.

3) I intuitively felt an important part of this study was mentioned in the discussion for the first time. Ln229-230: The results of your study suggest that increasing woody cover trends from multiple previous research articles are related to increasing crown size, rather than increasing density. This is huge and forms the central finding but is only mentioned once! There are important implications for global carbon cycles (see Poulter et al. 2014 Nature and Liu et al. 2015 Nature Climate Change), bush encroachment

etc. This would be a finding that other scientists would explore further. You need to develop this theme. I want to know more!

Response: We will further emphasize this finding by explaining implications for the global carbon cycle. We do not think, though, that our results implicate that bush encroachment in Africa results from increasing crown sizes instead of more woody plants.

4) PVPs: This aspect of your study is mentioned briefly in the beginning, forms a large chunk of your results and more than half of your discussion. This leaves the manuscript unbalanced and the reader is left wondering why PVPs are so important and why it was decided to explore it so heavily. If the focus is on PVPs, that needs to be reflected in the abstract (it isn't mentioned once until the end) and introduction (it is mentioned as a phenomenon but no why they are worth exploring or what the question is about them. While I am no specialist in PVPs, I would also suggest that no lit review section of PVPs is complete without a mention of Max Rietkerk's work, particularly Rietkerk & van de Koppel 2008 Trends in Ecology and Evolution. I was also missing mention of Bromley et al. 1997 Journal of Hydrology which specially mentions West African PVP's and 'tiger bands'.

Response: We agree with the reviewer comments and will modify the abstract to increase emphasis on PVPs and explain why PVPs are important to our understanding of drylands. We will also reference the work by Rietkerk.

Specific comments

1. Title: The title has the word "Savannas" in it. Yet, later on the authors mention 'drylands' (ln 40) which contains large areas not typically counted as savannas. In Figure 2 the vegetation area of interest is labelled 'rangelands', as well as in ln 87. Why use Ellis & Ramankutty's anthropogenic biome for an abiotic-vegetation study on savannas when you could use a climatic-disturbance based biome which defines savannas? This study does not consider the human component. Whichever term

the authors choose require clarifiAcation and should be used consistently. Why not provide a map of the savanna extent? For example, the 'rangeland' areas in Morocco, Algeria and Libya are traditionally not considered savannas. The authors could use the extent used by Sankaran et al. 2005 Nature as it is widely accepted. It could even be interesting to see the relative differences in abiotic influences on your sites divided into the "stable" and "unstable" savanna categories, if they agree with Sankaran et al. Just a thought. The other issue with the title is the word "structure" when your metrics measure woody cover and tree density. My understanding is that 'structure' implies height metrics or SCD's.

Response: We agree with the reviewer's worry that the original manuscript used too many terms: savannas, drylands, and rangelands. Most of the sites are savanna but there are also a few sites without trees in the dataset. We also have humid savanna sites so neither drylands nor savanna is a perfect match. In the revised manuscript we settle on 'savanna' as the most inclusive terminology. We used Ellis & Ramankutty's map as a guideline for sampling across African savannas and avoid sampling in the rainforest, agricultural and urban areas. We use the term "structure" describing tree density and crown sizes, i.e. the horizontal vegetation structure. While we are not able to analyze vegetation height, we nevertheless feel the terminology is appropriate. We are not alone in this interpretation as there are many studies on vegetation structure that do not analyze vertical structures.

2. Abstract: PVPs not mentioned until ln 28. They need to be introduced earlier if they are the focus of the study.

Response: We agree and will introduce PVPs earlier.

3. Ln 12-15 Very concise and clear summary of your introduction and aim in these abstract sentences. This idea also needs to be explicitly stated upfront in the introduction and well referenced. I probably lost the impact of this point in the introduction because of poor flow and structure.

Response: We agree and will modify the manuscript.

4. ln33-34 "While humans often play a dominant role in many systems. . ." I did not understand the point of this statement and it feels out of place here. Either remove it or expand on it.

Response: We remove this statement when updating the introduction.

5. ln 38 ". . .future stability and productivity. . ." 'stability is a loaded term in savanna literature. Perhaps rephrase this. This idea would form a nice link to bring up again in your conclusion to tie your manuscript together.

Response: Will re-phrase this to avoid the term "stability"

6. ln 44-45 Great to bring up fire's influence. Recent work by Smit et al. 2016 Journal of Applied Ecology show that SCD's are affected by high intensity fires, including tall tree (large canopy size?) loss. I understand that fire intensity can't be ascertained with MODIS data, but it does need to be mentioned that intensity plays a role.

Response: Fire intensity certainly matters and we will add acknowledgement of that in the sentence on line 44.

7. ln 40-50 This may be a personal style preference, but worth a mention (word limit permitting). The first half of the paragraph lists abiotic driver influences on woody veg properties, the 2nd half specifies how these drivers can influence the specific metrics of the study (individual: crown size; population: crown density, woody cover) and provide an example of how the same woody cover can have different ecosystem functioning. This is a natural flow, but I wanted a bit more on both topics. Could these two sections be expanded to their own paragraphs?

Response: We will take this into account when updating the introduction. Thanks for your suggestion!

8. ln 64-65. Please reference mention of vegetation bands. Rasmussen? Bromley?

Response: Will add reference.

9. ln 69-70. Both studies the authors cite for "African savannas" are from W Africa. Could other African studies be included?

Response: Will try to find a similar study from Southern or Eastern Africa.

10. ln +-73-81 This paragraph seems more suited to the methods section. Perhaps you could reduce these details to a sentence or two, linking the methodological processes to the general aim, rather than mention details here and then details again in the very next paragraph? Figure 1 should also only be mentioned in the methods.

Response: We agree with these comments and will update accordingly.

11. ln 80 The PVPs identified in the study sites, were those sites derived from the literature or were they found by the authors. Please mention this. If the latter, it would be nice for the reader to have image examples of the different kinds of PVPs. Are they very easy to spot?

Response: They were identified visually by the authors (ln 184-185). We agree it would be clarifying to include images of PVPs and take note of this suggestion.

12. ln 90. Does this mean spring in the northern and southern hemispheres? Could you be more specific?

Response: We have added wording to the manuscript to explain the approach here. "... when trees were in full leaf (generally in mid to late growing season)"

13. ln 91-92 It's not necessary to mention that another on-going study influenced this one's parameters unless some of the data from that study are included in this paper. Perhaps leave this out.

Response: We understand the reviewer's comment that it may not be necessary to reference the following study, however, in the end we decided to keep this wording since it impacts site selection shown in Figure 2.

[Figure]

14. Methods: The sections on preprocessing and classification were thorough. Thanks!

Response: Thank you! 15. In 112-115 This section isn't really necessary for the article, although I do understand the feeling of wanting the time and monumental effort taken for analyses to be recognised by the readers!

Response: We will remove the last two sentences of this paragraph as suggested.

16. ln 132 Is there a reason for the 40 m limitation to crown size?

Response: We set the upper limit to crown-size at 40m as we thought larger trees would be very rare across our entire sample domain. In reality, the delineation process was very rarely (if ever) affected by this rule as the crown merging procedure seldom resulted in crowns of that size.

17. lns 143-154 This is a well-needed section and I like that the limitations are mentioned. However, it needs bolstering with supporting literature. A quick google search has shown that crown delineation techniques with multispectral, high resolution satellite data exist and it would useful to see a comparison of the trade-offs to back up the method you have used. This ties in with my request to see support for the NDVI crown centre identiïfication method. Accuracy statistics would be a useful addition here.

Response: We agree and will add references to this section. We will also refer to results from the added validation/error Appendix.

18. ln 172 Was a 20 m cut-off used for Ripley's L because that is where the sill occurs on all the curves in Figure 5?

Response: We chose to evaluate L at 20 m to be at length scales coarser than typical savanna tree crown diameters, and within length-scales of facilitative tree-tree effects. This distance also makes sense from observation of the sills in Figure 7.

19. ln 192 Mentions Figure 4. Figure 3 was never mentioned. Please include it where

relevant.

Response: Will add reference to Figure 4. Thank you.

20. Results: The subheadings seem strange. You have one sentence on vegetation characteristic differences followed by a subheading "3.1. Mean crown size, density and woody cover". Surely the previous paragraph (of one sentence) fits into this subheading? Or was the subheading meant to be related only to BRT results?

Response: We include the short paragraph at the beginning of the Results section to introduce the reader to Figure 4. Since Figure 4 also contains aggregation we preferred not to include it in Section 3.1

21. In 194 It was not clear to me from Figure 4 that arid sites had higher levels of aggregation. Perhaps because the colours did not come out well in that panel?

Response: The three curves overlap considerably for the L-function. However, we infer higher aggregation in the drier systems. The wet sites (>700) have the highest peak close to zero, while the arid sites (<400) have a lower peak at zero and is more spread out over a wider range of values.

22. In 197 "Woody cover and mean crown size both had strong relationships with the local environment. . ." What factors in the local environment? Could you be more specific?

Response: We will rephrase this. We were referring to the higher R-squared in general when mentioning strong relationships with the local environment.

23. In209-211 Nice findings. The sentence that starts " These are factors that in-fluence ecohydrological processes. . .."at the end of the paragraph is better suited to the discussion section and needs to be referenced.

Response: Agreed. We will merge this text into the discussion section.

24. In 219. ". . .aggregation reaching a minimum at around 25 meters." Consistency

with meters/m. This sentence also needs a figure reference at the end. Figure 7?

Response: We will use m instead of meters, and add figure reference.

25. ln 219-220. This is a discussion point.

Response: Will move to the discussion.

26. "Heading 4.1. Dividing woody cover into density and crown size components" as well as ln 226-228 are concepts that should be addressed in the introduction. This is a key part of what makes this study novel as most research deals with woody cover without addressing density/crown size differences. These lines are the coherent aim and motivation I was missing in the introduction.

Response: We will modify the Introduction so that this point is more clearly made.

27. ln 229-231 Great finding! Make a meal of it. The authors need to discuss this vs. bush encroachment findings in the literature.

Response: We will emphasize the novel finding that increasing cover is a function of tree size more than tree density. We are not sure of the reviewer's point relating to bush encroachment so for now have not addressed this suggestion.

28. ln 243-245 Low woody cover unrelated to rainfall seasonality. This section needs mention of the large role of disturbance agents in "unstable savannas" (sensu Sankaran et al. 2005). The authors do mention elephant impacts in a sentence, but this needs more unpacking and forms part of the caveats to this study's results as biotic disturbance was not included. Together with acknowledging effects of fire intensity on SCDs.

Response: We will mention herbivory and other influential factors not captured by the analysis.

29. ln 254 "In accordance with previous literature. . ." There are no references at the end of this sentence. Which literature?

Response: Will add references.

30. ln 270 ". . .and short-range facilitation through modified microclimate close to nursery plants" needs a reference.

Response: Will add reference.

Technical corrections

1. ln20-30 Be careful of the change in tenses. Generally, methods and results should be reported in the past tense.

Response: We will pay more attention to changes in tenses. Good point!

2. Journal editor preference, but Figure mentions should normally be in parentheses, rather than mentioned in the sentence. Eg. "Frequency distributions of the four woody properties, separated into three rainfall categories, are shown in Figure 4." To "The more arid savannas (<400 mm/year) typically feature smaller crown sizes, lower crown density and woody cover, and higher levels of aggregation than sites in the wetter categories (Figure 4)."

Response: Will modify text.

3. ln 118 Insert spaces between "240x240" and shouldn't 'meter' be 'm'. Be consistent throughout the manuscript. Either change previous mention of 'meter' to 'm' or vice versa.

Response: Will change to m and add spaces

4. ln 124 'ID' or 'point' rather than 'id'

Response: Will change to ID

5. ln 241. This is the only occasion a discussion sentence refers to a results Figure. Either include more links to the results where appropriate, or remove this one. Consistency. e.g. ln 228-229 could also use a figure reference?

Response: Will add more links in the discussion.

---

## Referee Comment (RC2) · S. de Bruin (Referee) · 10 Apr 2017

This work reports on the preparation of a sample of 876 records with woody vegetation properties and an analysis of differences in those properties in response to environmental variation in terms of soil characteristics, rainfall quantity and distribution and slope. The authors are to be commended for their great effort of generating a considerable and interesting data set based on high resoution image data using a generally well-described methodology. It would be great if the data were made available to other scientists.

I am less convinced of the rigor of the subsequent analyses, though. Given my background, I have focussed my review on methodological issues.

Specific comments:

1) Section 2.1 of the paper should include a proper definition of the sampling universe as well as a description of the sampling frame. The section lists sampling criteria, but these seem to address a pragmatic approach for dealing with issues that occurred while preparing the data set rather than a design approach targeting the intended population.

2) Methods section 2.3 (Crown delineation) contains discussion (lines 135-139 and 144-146) which is improperly placed in my view. The methods section should just describe the methods, as used. Alternative methods can be described in the introduction while potential flaws in the results caused by the used methods should be described in the discussion section.

3) Same section (lines 148-150): Is it really enough to balance rates of falsely divided and falsely grouped crowns? I guess one wants to minimize those errors. How was this achieved?

4) Same section (lines 150-151) The authors seem to have validated the results by visual inspection which showed the results to "look realistic". That is by no means a scientific validation!

5) The validation exercise described in the appendix concerns a small dataset in Kenya. In the sample, common large umbrella thorn acacias were claimed to be overrepresented and given their problematic behaviour in determining crown size and crown density they were excluded when computing R2. So, how can the results from this exercise be generalized to the entire dataset?

6) It remains unclear to me how vegetation periodicity was characterised. In line 185 (section 2.5), "spotted, labyrinthine, gapped or banded patterns" are briefly referred to (between brackets). This seems to suggest periodicity was identified on a single image. Since periodic behaviour plays an important role in the analyses and conclusions, it is necessary to explicitly describe whether or not multiple images were used and to be be very clear on its characterisation.

7) The analysis employs a mix of resolutions (support sizes) but I am unsure on how these were integrated. It is mentioned that the TRMM data were resampled by bilinear interpolation, but for the other data sets it remains unclear to me at what resolution the analyses were performed. For example, were average slopes over the 240 x 240 m2 regions used or were patterns within the 240 m cells also considered?

8) There are several changes of tenses throughout the text (also mentioned in the review of Penny Mograbi). My understanding is that the present tense is reserved for presenting either well-known facts or statistical inferences from sample statistics that are generalised to entire populations. However, in this paper no formal hypothesis testing is performed; all results should thus be in past tense since they concern the used (sample) data set.

9) The previous comment points to a major weakness of the work: One might doubt whether the analyses support drawing general conclusions about woody vegetation properties in response to environmental variation in African savannas. The sampling method would only allow to do so under the assumption that the sample is representative. This should then be explicitly stated and supported by proper argumentation. Furthermore, at some places the authors acknowledge that the used data are not error free. This implies that we are uncertain about the true environmental properties and the woody vegetation characteristics. The question then arrises whether the observed differences or relationships exceed uncertainty bounds. How did the authors decide whether an effect was "clear positive", "weak" or "absent"? The inference mechanism should be described.

10) The grey dots in Figures 5 and 6 are claimed to represent fitted values for each of the 876 sites considering a single environmental variable with the other variables fixed at their avarages. For MAP, rain seasonality, sand content and slope this seems to indicate erratic behaviour at very minor changes of the environmental variable under consideration. For "fire frequency" a vertical banding pattern is observed which suggests the BRT produced multiple outputs for the same fire frequency. How come? This

pattern should be explained!

---

## Author Comment (AC3) · 21 Apr 2017

We thank the reviewer for constructive feedback on the methodology. Here are our responses:

Specific comments:

1) Section 2.1 of the paper should include a proper definition of the sampling universe as well as a description of the sampling frame. The section lists sampling criteria, but these seem to address a pragmatic approach for dealing with issues that occurred while preparing the data set rather than a design approach targeting the intended population.

Response: We have modified the manuscript to clarify that the sampling frame for the

analysis was sub-Saharan African savannas with a minimum of anthropogenic distur-bances. We also added that within-image site-selection followed a systematic sampling approach and was guided by a 0.04° longitude/latitude grid.

2) Methods section 2.3 (Crown delineation) contains discussion (lines 135-139 and 144-146) which is improperly placed in my view. The methods section should just de-scribe the methods, as used. Alternative methods can be described in the introduction while potential flaws in the results caused by the used methods should be described in the discussion section.

Response: These lines describe how this delineation method relates to previous de-lineation approaches, and concerns related to this and other delineation methods. As the purpose is to describe the method and its strengths/weaknesses, we do not think these sentences are inappropriate for the Methods chapter. We understand the re-viewer's concerns, but feel that it is better to keep the description of this method in a single section instead of dividing it between Methods and Discussion.

3) Same section (lines 148-150): Is it really enough to balance rates of falsely divided and falsely grouped crowns? I guess one wants to minimize those errors. How was this achieved?

Response: This is a general statement about the consequences if crowns are system-atically falsely divided or falsely merged, which is an issue with all crown delineation methods. Originally, fine-tuning of the delineation method was done by visual inspec-tion of the crown polygons overlaying the high resolution imagery. With the added Appendix, we also refer to the validation of the Kenyan sites.

4) Same section (lines 150-151) The authors seem to have validated the results by visual inspection which showed the results to "look realistic". That is by no means a scientific validation!

Response: Here we will refer to the validation of the Kenyan sites, which is a quantitative validation. The visual inspection does, however, also play a role since it helped us determine that the delineation was consistently executed across all landscapes. In many scenes, individual trees can be identified from visual inspection.

5) The validation exercise described in the appendix concerns a small dataset in Kenya. In the sample, common large umbrella thorn acacias were claimed to be overrepresented and given their problematic behaviour in determining crown size and crown density they were excluded when computing R2. So, how can the results from this exercise be generalized to the entire dataset?

Response: The large majority of our sites do not contain this type of trees with particularly large and spread out crowns which are relatively rare across all of African savannas. Since all four sites in Amboseli were dominated by them, we determined they were overrepresented in the field data. We acknowledge that the delineation method underestimates crown sizes for trees with large spread out crowns, and will mention this problem when referring to the appendix.

6) It remains unclear to me how vegetation periodicity was characterised. In line 185 (section 2.5), "spotted, labyrinthine, gapped or banded patterns" are briefly referred to (between brackets). This seems to suggest periodicity was identified on a single image. Since periodic behaviour plays an important role in the analyses and conclusions, it is necessary to explicitly describe whether or not multiple images were used and to be very clear on its characterisation.

Response: We will clarify how sites with periodic patterns were identified. We have also added images of sites with periodic patterns, as suggested by the previous reviewer. We visually inspected each site individually and determined if it had a periodic vegetation pattern. This is straightforward for clear cases of banded and spotted patterns. There were cases where it was less straightforward, e.g. weak gapped patterns, and for those we tried to be consistent with the assessment.

7) The analysis employs a mix of resolutions (support sizes) but I am unsure on how

these were integrated. It is mentioned that the TRMM data were resampled by bilinear interpolation, but for the other data sets it remains unclear to me at what resolution the analyses were performed. For example, were average slopes over the 240 x 240 m2 regions used or were patterns within the 240 m cells also considered?

Response: We only considered the center point of the sites and extracted raster values based on nearest neighbor in all cases except the TRMM where we used bilinear interpolation. We will clarify this in the text.

8) There are several changes of tenses throughout the text (also mentioned in the review of Penny Mograbi). My understanding is that the present tense is reserved for presenting either well-known facts or statistical inferences from sample statistics that are generalised to entire populations. However, in this paper no formal hypothesis testing is performed; all results should thus be in past tense since they concern the used (sample) data set.

Response: We have modified the text to correct tenses.

9) The previous comment points to a major weakness of the work: One might doubt whether the analyses support drawing general conclusions about woody vegetation properties in response to environmental variation in African savannas. The sampling method would only allow to do so under the assumption that the sample is representative. This should then be explicitly stated and supported by proper argumentation. Furthermore, at some places the authors acknowledge that the used data are not error free. This implies that we are uncertain about the true environmental properties and the woody vegetation characteristics. The question then arrises whether the observed differences or relationships exceed uncertainty bounds. How did the authors decide whether an effect was "clear positive", "weak" or "absent"? The inference mechanism should be described.

Response: While the sample set was affected by various factors, including image availability, cloud cover in images, and anthropogenic disturbances, we do not see any of

these factors creating a bias when relating the woody structure estimates to environmental variables. The inferred "clear positive" or "weak", relationships were based on the trends in the partial dependence plots. We agree that interpretation of results from boosted regression trees relies to some extent on a qualitative assessment of these trends. When describing the methodology we have added: "The influence of individual predictors was estimated from their relative importance in the BRT models, and the directions of relationships were inferred from their trends in partial dependence plots."

10) The grey dots in Figures 5 and 6 are claimed to represent fitted values for each of the 876 sites considering a single environmental variable with the other variables fixed at their avarages. For MAP, rain seasonality, sand content and slope this seems to indicate erratic behaviour at very minor changes of the environmental variable under consideration. For "fire frequency" a vertical banding pattern is observed which suggests the BRT produced multiple outputs for the same fire frequency. How come? This pattern should be explained!

Response: Only the partial dependencies (red lines) account for the average effect of the other variables. We will clarify the text and avoid the term fitted function since it might cause confusion. The erratic behavior (overfitting the data points) is often seen in partial dependency plots of boosted regression trees and is perhaps a weakness of this method. It means we need to focus at the main trend of the response function. Many sites had the same fire frequency (based on the number of fires in the period 2001-2015) which causes the striped pattern. The fitted values are model predictions based on all predictors and will thus vary. We have updated the text to clarify this.

---

## Author Comment (AC4) · 1 May 2017

We have made updates to the manuscript in accordance with reviews and comments. I attach a version of the updated manuscript with tracked changes relative to the original version. We have not responded to the short comments that came in during the discussion but are grateful for all feedback and have used it to improve the manuscript.

Please also note the supplement to this comment:
http://www.biogeosciences-discuss.net/bg-2017-7/bg-2017-7-AC4-supplement.pdf

**Supplement:**

**Patterns in Woody Vegetation Structure across African Savannas**

Christoffer R. Axelsson[1] and Niall P. Hanan[2]

[1]Geospatial Sciences Center of Excellence, South Dakota State University, Brookings, SD, USA

[2]Plant and Environmental Sciences, New Mexico State University, Las Cruces, NM, USA

*Correspondence to*: Christoffer R. Axelsson (christoffer.axelsson@sdstate.edu)

Key words: African savannas, vegetation structure, environmental gradients, tree crown delineation, crown size, crown density, tree aggregation, woody cover, rainfall seasonality, periodic vegetation patterns, patchiness

**Abstract.** Vegetation structure in water-limited systems is to a large degree controlled by ecohydrological processes, including mean annual precipitation (MAP) modulated by the characteristics of precipitation and geomorphology that collectively determine how rainfall is distributed vertically into soils or horizontally in the landscape. We anticipate that woody canopy cover, crown density, crown size, and the spatial distribution of woody plants in the landscape, will vary across environmental gradients. Periodic vegetation patterns (PVPs) is an extreme form of vegetation aggregation that is attributed to ecohydrological processes and 
[revised manuscript text omitted]

---

## Author Comment (AC5) · 17 May 2017

We thank Susan Walsh for the comment:

The authors state: "Because of uncertainty in the accuracy of the woody properties derived from the delineated crowns, we do not focus on absolute numbers but on how they vary across environmental gradients." The study assumes that errors in the woody proportion estimation are either negligible or are the same across the environmental gradients and at the different African sites. What evidence is there for this? Tree crown morphology (roundness etc.) and remote sensing signal contributions (i.e., soil, under-story and shadow) are unlikely to be the same across the African sites/gradients and so errors in the empirical crown delineation method are likely to be variable. Moreover, the tree crown delineation is applied to different satellite data (WorldView, GeoEye,

Quickbird).

Without a quantitative accuracy assessment of the tree crown delineation across the gradients/sites and among the data used this interesting study is incomplete and potentially flawed - the results may be controlled by crown delineation differences.

Response: The added appendix with field data from Kenya provides an estimate of the accuracy of the crown delineation methodology. As you say, there is a variation in terms of soil background, shadowing, tree morphology etc. across the different sites. The flexible classification approach, with manual handling of all sites, helped us adjust to each site individually and made sure that the final classified images appeared correct from a visual inspection. A fully automatic classification approach would not have achieved this due to variability in the mentioned factors.

It was not possible for us to use data from a single satellite due to limited availability. There is variation in the properties of the imagery, both due to different satellites and due to variation in solar and viewing angles. The orthorectification, resampling to 0.6 m ground resolution, and the flexible classification approach served to minimize the effect of varying image properties in the final classified results.

---

## Author Comment (AC6) · 17 May 2017

Christoffer R. Axelsson and Niall P. Hanan

christoffe@hotmail.com

We thank Aoife Hutton for comments on the Discussion section:

1. In Discussion, the relationship between woody cover, crown size and crown density is introduced (Line 225), yet seems not to be mentioned in any section before – perhaps it would be fitting to highlight this relationship in an earlier section.

Response: We have re-arranged this and it is now introduced in the Introduction.

2. In line 228, the term 'woody density' is used, causing the reader some confusion, as previously the distinction was made between woody cover and woody density. Is this new term intentional?

Response: We have updated the manuscript to use the term "crown density" consistently and avoid "woody density".

3. We had some queries surrounding the focus on sites with rainfall seasonality above 0.8 (Line 245) - what is the justification for this focus? Is this where a sensitivity lies in terms of woody density response, and is this a causative relationship? Some more detail here would benefit the discussion.

Response: We mentioned 0.8 to draw attention to the linear relationship that appear if the less seasonal sites in East Africa are excluded. The East African sites appeared to diverge from the general pattern and we suspected that this could be a result of elephant browsing and not rainfall seasonality.

4. On line 243, 'previous literature' is mentioned but no references are given immediately. Although individual associations are then explained and cited, the opening sentence omitting these references could be improved by restructuring this paragraph.

Response: We have added a reference to the opening sentence in this paragraph.

5. The use of the word 'So' to open line 268 does engage the reader, but seems inconsistent with the more formal writing style in the rest of the paper. General comments on the study Ecosystem services is mentioned near the beginning of the paper but is not revisited in discussion – how relevant are ecosystem services to this study? 'Processes' seem like a more accurate description of relationships mentioned. Although the effects of primary variables were investigated, no interaction effects investigated. Are there any coupling effects of say, rainfall and slope on woody density?

Response: Agreed, "So" has been removed. Ecosystem services is not a focus of this paper, but served as an example for why it is important to know about tree sizes and densities as opposed to only know the woody cover. The boosted regression tree R package identifies interactions between variables, but we chose not to report them. We deemed interactions were less important than the main response functions and it would have taken up much space to visualize all interactions.

---

## Author Comment (AC7) · 17 May 2017

We thank Gemma Kitson for comments on the Methods section:

The last paragraph of the introduction seems to go into rather a lot of detail, and some of the things discussed in it could perhaps have been kept for the methods section.

Response: We have moved parts of the last paragraph in the Introduction to the Methods section.

The 'initial unsupervised classification with manual assignment into woody, herbaceous, and bare cover classes' is rather vague – it is not clear who was carrying out the classification or in what way they were unsupervised.

Response: Unsupervised classification is a type of classification technique. We believe

most readers are familiar with this term.

It would be useful to include the number of sites that were initially considered for inclusion in the study and the number that were eliminated, perhaps before stating the final number of sites, to make it clearer that the value of 876 is after all sites not suitable for inclusion have been eliminated.

Response: There were actually several reasons for why we abandoned certain sites including: cloud cover, unknown processing problems in ENVI at specific sites, clear anthropogenic disturbances, low image quality, few green leaves on trees, and difficulty in separating trees from grasses. We felt that it was of minor importance to explain in detail how many sites were excluded due to each of these factors.

Some parts of the methods section lack justification for the decisions that were taken. For example, in line 132 the authors state that segments were not merged if the resulting crown size has a diameter greater than 40m, but there is no justification for why this particular value has been chosen, making it seem as if it is at random.

Response: We have changed the sentence on line 132 to clarify this.

In line 133, the authors state that they experimented with different methods before settling on the ones they actually used, but do not explain why they didn't use the previous methods; without this justification, mentioning that alternative methods were decided against seems to provide no insight.

Response: We deleted the sentence on line 133.

It would be interesting to know how long the image classification process took, since the authors indicate that it was the most time-consuming part of the analysis – without knowing how long the analysis took in total this piece of information doesn't tell the reader very much, although it is interesting to know.

Response: We deleted the sentence about it being the most time-consuming part of the analysis.

An estimation of the accuracy of the methods would be useful, especially concerning the delineation of tree crowns. It is possible there might be a bias in the estimation of crown size across environmental gradients, meaning that the rate of falsely divided and falsely grouped crowns may be different at different values of environmental factors. If this is the case, even comparing across environmental gradients could be inaccurate. In line 150 the authors state that the generated crown layers 'look realistic from a visual inspection across all landscape types and different tree densities', but this is a very qualitative way of assessing the accuracy of the methods; a quantitative value of accuracy would be better.

Response: We have added an appendix with a quantitative accuracy analysis using field data from Kenya.

---

## Author Comment (AC8) · 17 May 2017

We thank Galina Toteva for comments on the Abstract and Results sections:

Abstract: The abstract gives a good overview of the scope of the paper. However, the same amount of information could probably be conveyed with fewer words. Additionally, the description of the results is particularly detailed but not quantitative enough. Finally, the abstract would benefit from concluding sentence.

Response: We added numbers for change in woody properties with increasing rainfall to make it more quantitative. We also deleted a sentence that was not needed.

Results section: You mentioned that there was a "strong relationship" between woody cover and mean crown size (line 197). However, you did not state any threshold in

regards to that value or how you arrived at it. Similarly, you referred to the "large majority" of sites (line 206) without specifying exactly how many. Another suggestion that we have about the results section, is to consider the interaction effect between the variables in the analysis. Generally, it is not entirely clear whether you answered your research question and how you can quantify the success of this research project. It is also more convenient to have the graphs and figures within the text rather than in an appendix but we assumed this is because the paper is still undergoing review.

Response: We have rephrased the paragraph starting on line 197 to clarify it. We also added a number for the percentage of sites on line 206 to make it less vague.

---

## Author Comment (AC9) · 17 May 2017

We thank Elizabeth Hanlon for comments on the Introduction and Conclusion sections:

Introduction

1. On line 35 the authors state "climate, both rainfall patterns and temperatures, could change in many parts of Africa" and then the reference. It would benefit the reader if some of these changes, as they are relevant to your research, were stated and the effect that these changes could have on vegetation.

Response: We chose not to delve deeper into describing climate change scenarios because it is not a focus of this paper.

2. The use of words like 'these' and 'those' should be avoided to remove any ambiguity

about the subject of the statement.

Response: We have updated the manuscript but several occurrences of "these" remain. We do not see any reason for ambiguity among the remaining cases.

3. The end sentence in the second introduction paragraph (beginning line 51) could be improved to reduce vagueness; how do we learn about the impacts of the underlying ecosystem processes?

Response: We have re-written and re-arranged this part of the Introduction. The sentence on line 51 was removed.

4. Aggregation is discussed a lot but there is relatively little introduction to it. The relationship between woody plants and aggregation should be given some context.

Response: We have added a few sentences to give aggregation and PVPs more context.

5. A diagram of the spotted, labyrinthine, or gapped patterns of PVPs would be useful for readers less familiar with PVPs.

Response: We added a figure with images of PVPs.

6. The last paragraph of the introduction seems out of place, and would be better suited for the methods section. A smaller summary of your work would be appropriate for the introduction, and Figure 1 would definitely be better placed in the method section.

Response: We have modified the last paragraph and removed two sentences that described the methodology. Figure 1 is in the Methods section.

7. It would be better to end the introduction with a research question, or the aim of your experiment. A lack of clear hypotheses made it hard to read the results and judge whether the experiment was successful or not. A clear research question helps the reader know what you are trying to achieve.

Response: The research question "how do woody cover, crown size, crown density and the spatial pattern of trees vary with environmental gradients" is stated in the beginning of the last paragraph of the Introduction.

Conclusion

1. The conclusion was more of a summary of your results, rather than a rounding up of the issue explored. Ideas for future work could be given, and the importance of the work restated.

2. The phrase "possible difference maker" is a clumsy end to the paper.

Response: We made some minor modifications to the Conclusion section to try to improve it. We changed "possible difference maker between" to "possible difference between".